# Representation Norm Amplification for Out-of-Distribution Detection in Long-Tail Learning

**Dong Geun Shin**  *dgshin@kaist.ac.kr*
*School of Electrical Engineering*
*Korea Advanced Institute of Science and Technology (KAIST)*

**Hye Won Chung**  *hwchung@kaist.ac.kr*
*School of Electrical Engineering*
*Korea Advanced Institute of Science and Technology (KAIST)*

**Reviewed on OpenReview:** *https://openreview.net/forum?id=z4b4WfvooX*

## Abstract

Detecting out-of-distribution (OOD) samples is a critical task for reliable machine learning. However, it becomes particularly challenging when the models are trained on long-tailed datasets, as the models often struggle to distinguish tail-class in-distribution samples from OOD samples. We examine the main challenges in this problem by identifying the trade-offs between OOD detection and in-distribution (ID) classification, faced by existing methods. We then introduce our method, called *Representation Norm Amplification* (RNA), which solves this challenge by decoupling the two problems. The main idea is to use the norm of the representation as a new dimension for OOD detection, and to develop a training method that generates a noticeable discrepancy in the representation norm between ID and OOD data, while not perturbing the feature learning for ID classification. Our experiments show that RNA achieves superior performance in both OOD detection and classification compared to the state-of-the-art methods, by 1.70% and 9.46% in FPR95 and 2.43% and 6.87% in classification accuracy on CIFAR10-LT and ImageNet-LT, respectively. The code for this work is available at `https://github.com/dgshin21/RNA`.

## 1 Introduction

The issue of overconfidence in machine learning has received significant attention due to its potential to produce unreliable or even harmful decisions. One common case when the overconfidence can be harmful is when the model is presented with inputs that are outside its training distribution, also known as out-of-distribution (OOD) samples. To address this problem, OOD detection, i.e., the task of identifying inputs outside the training distribution, has become an important area of research (Nguyen et al., 2015; Bendale & Boult, 2016; Hendrycks & Gimpel, 2017). Among the notable approaches is Outlier Exposure (OE) (Hendrycks et al., 2019a), which uses an auxiliary dataset as an OOD training set and regulates the model's confidence on that dataset by regularizing cross-entropy loss (Hendrycks et al., 2019a), free energy (Liu et al., 2020), or total variance loss (Papadopoulos et al., 2021). These methods have proven effective and become the state-of-the-art in OOD detection. However, recent findings have highlighted the new challenges in OOD detection posed by long-tailed datasets, characterized by class imbalance, which are common in practice. Even OE and other existing techniques struggle to distinguish tail-class in-distribution samples from OOD samples in this scenario (Wang et al., 2022b). Thus, improved methods are needed to tackle this new challenge.

To address OOD detection in long-tail learning, we need to solve two challenging problems: (i) achieving high classification accuracy on balanced test sets, and (ii) distinguishing OOD samples from both general and tail-class in-distribution (ID) data, which is underrepresented in the training set. While previous works have explored each problem separately, simply combining the long-tailed recognition (LTR) methods (Kang et al.,

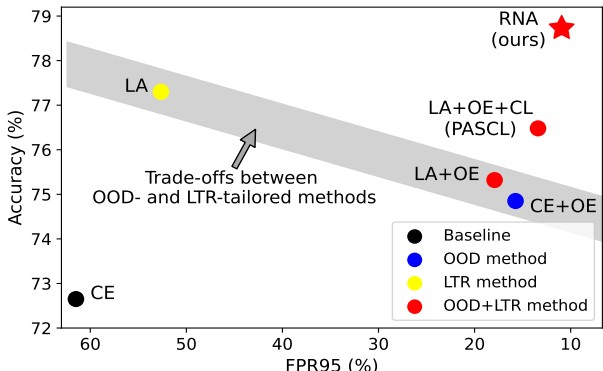

Figure 1: The OOD detection performance (FPR95) and classification accuracy (Accuracy) of models trained by various methods. OE (Hendrycks et al., 2019a), designed for OOD detection, and LA (Menon et al., 2021), aimed at long-tailed recognition (LTR), exhibit trade-offs between OOD detection and ID classification when combined (LA+OE). In contrast, our method (RNA) excels in both FPR95 and Accuracy, effectively overcoming these trade-offs.

2020; Menon et al., 2021) with OOD methods does not effectively solve this challenge (Wang et al., 2022b). This is because the existing LTR and OOD methods often pursue conflicting goals in the logit space, creating trade-offs between OOD detection and classification, especially for tail classes (see Figure 1 and Table 1). For example, OE (Hendrycks et al., 2019a) uses auxiliary OOD samples to train the model to output uniform logits for rare samples, while the LTR method such as Logit Adjustment (LA) encourages a relatively large margin between the logits of tail versus head classes. Combining these approaches results in better OOD detection but inferior classification compared to LTR-tailored methods, while achieving worse OOD detection but better classification than OOD-tailored methods. The crucial question is: how can we achieve both goals without compromising each other?

To address this challenge, we provide a new OOD detection method in long-tail learning, called Representation Norm Amplification (RNA), which can effectively decouple the classification and OOD detection problems, by performing the classification in the logit space and the OOD detection in the embedding space. Our proposed training method creates a distinction between ID and OOD samples based on the norm of their representations, which serves as our OOD scoring method. The main idea is to train the classifier to minimize classification loss only for ID samples, regularized by a loss enlarging the norm of ID representations. Meanwhile, we pass auxiliary OOD samples through the network to regularize the Batch Normalization (BN) (Ioffe & Szegedy, 2015) layers using both ID and OOD data (Figure 3a). This simple yet effective training method generates a discernible difference in the activation ratio at the last ReLU layer (before the classifier) and the representation norm between ID vs. OOD data, which proves valuable for OOD detection. In addition, since the loss only involves ID data, only ID data is involved in the gradient for updating model parameters. This focus on learning representations for classification, rather than controlling network output for OOD data, is a key aspect in achieving high classification accuracy. Our contributions can be summarized as follows:

- We introduce Representation Norm Amplification (RNA), a novel training method that successfully disentangles ID classification and OOD detection in long-tail learning. RNA achieves this by intentionally inducing a noticeable difference in the representation norm between ID and OOD data. This approach enables effective OOD detection without compromising the models' capability to learn representations for achieving high classification accuracy.

- We evaluate RNA on diverse OOD detection benchmarks and show that it improves FPR95 by 1.70% and 9.46% and classification accuracy by 2.43% and 6.87% on CIFAR10-LT and ImageNet-LT, respectively, compared to the state-of-the-art methods without fine-tuning.

## 2 Related works

**OOD detection in long-tail learning**   While substantial research has been conducted in OOD detection and long-tail learning independently, the investigation of their combined task, referred to as LT-OOD, has emerged only recently. The LT-OOD task encompasses three main lines of research: training methodologies (Wang et al., 2022b; Choi et al., 2023), post-hoc scoring techniques (Jiang et al., 2023), and abstention class learning (Miao et al., 2023; Wei et al., 2023) for LT-OOD tasks.

PASCL (Wang et al., 2022b) first addressed OOD detection in the context of long-tail learning, introducing a contrastive learning idea to improve the separability between ID tail classes and OOD data. BEL (Choi et al., 2023) tackled the LT-OOD task, highlighting class imbalances in the distribution of auxiliary OOD set. They proposed a method to regularize auxiliary OOD samples from majority classes more heavily during energy-based regularization. However, recent methods like PASCL and BEL employ the two-branch technique, where each branch is tailored for OOD detection and classification, respectively. In contrast, our method does not use such two-branch technique. Instead, we effectively address both challenges through a single model, leveraging the idea to disentangle ID classification and OOD detection, as will be observed in Section 6.2.

Jiang et al. (2023) tackled the LT-OOD task using a post-hoc method, adapting traditional OOD scoring to the long-tailed setting. Despite its effectiveness, this approach relies on model bias induced by class imbalances in the training dataset. Thus, it is hard to be combined with training methodologies aimed at resolving class imbalances to improve the classification accuracy.

Recently, abstention class-based methods tailored for LT-OOD tasks have emerged, including EAT (Wei et al., 2023) and COCL (Miao et al., 2023). EAT integrats four techniques: multiple abstention classes, CutMix (Yun et al., 2019) augmentation for tail-class ID data, the auxiliary branch fine-tuning, and ensemble models. COCL combines two techniques: debiased large margin learning and outlier-class-aware logit calibration. Debiased large margin learning entails training networks with two contrastive loss terms: one for tail-class ID prototypes and OOD samples, and the other for head-class ID samples and OOD samples. While these methods offer substantial improvements, they require a comprehensive approach with multiple techniques, and face challenges with OOD representations collapsing into the same class despite semantic differences.

**Norm-based OOD detection**   Previous studies have investigated the utilization of norms for OOD detection scores. Dhamija et al. (2018) pioneered the use of representation norms for OOD detection, with a training method, Objectosphere, directly attenuating representation norms of auxiliary OOD data during training to increase separation between ID and OOD data in the feature space. However, training with RNA loss proves more effective than directly attenuating OOD representation norms to reduce activation ratios and representation norms (see Section 6.3.1). CSI (Tack et al., 2020) incorporated norms into their scoring method, observing that the gap in representation norms between ID and OOD data merges implicitly in contrastive learning. NAN (Park et al., 2023) and Block Selection (Yu et al., 2023) also utilized feature norms for OOD detection, considering deactivated feature coordinates and selecting the most suitable hidden layer for norm utilization, respectively. While these approaches leverage representation norms for OOD scoring, they lack a training method aimed at widening the gap between the norms of ID and OOD data. In contrast, our method not only advocates for using representation norms for OOD detection in long-tail context but also propose a corresponding traning method.

There also exist techniques to address overconfidence issue for ID data by controlling the norm of logits. ODIN (Liang et al., 2018) uses temperature scaling for softmax scores at test time and LogitNorm (Wei et al., 2022) normalizes logits of all samples during training. However, these methods do not consider the underconfidence issue of tail classes. In long-tail learning, networks often produce underconfident outputs for tail predictions. Thus, solving both the overconfidence problem of OOD data and the underconfidence problem of tail ID data is crucial. Our method tackles this challenge by amplifying the norms of only ID representations, while indirectly reducing the representation norms of OOD data by BN statistics updates using ID and OOD data. More reviews of related works are available in Appendix A.

## 3 Background

### 3.1 OOD detection

In this section, we formally define the task of OOD detection. Consider a classification model that takes images as input and outputs predicted labels, consisting of a representation extractor $f_\theta$ with learnable parameters $\theta$ and a linear classifier $W$ (see Figure 3a). The model is trained on a labeled dataset $\{(x_i, y_i)\}_i$ to optimize $\theta$ such that $f_\theta(x_i) = y_i$ for all $i$. In-distribution data is sampled from the training data distribution, while out-of-distribution data comes from a different distribution. The main challenge of OOD detection task is that overparameterized deep neural networks tend to be overconfident in their predictions for OOD data. This overconfidence hinders the model's ability to distinguish between ID and OOD samples.

OOD samples are detected using a decision function $G$, which employs an OOD scoring function $S$. The function $S$ assigns a scalar score to each sample, indicating how the sample is likely to be an OOD data. The decision function $G$ is defined as:

$$G(x; \theta, W) = \begin{cases} \text{ID} & \text{if } S(x; \theta, W) < \xi \\ \text{OOD} & \text{if } S(x; \theta, W) \geq \xi \end{cases}, \tag{1}$$

where $\xi$ is a threshold.

Outlier Exposure (OE) (Hendrycks et al., 2019a) is an effective training methodology for OOD detection, using an auxiliary OOD set. The OOD data in this set are from a distribution disjoint with the distribution of both ID and the test OOD data. OE trains the model as regulating the output softmax vectors of auxiliary OOD samples to have a uniform distribution over classes. The detailed loss function is explained in Section 5.1.

### 3.2 Long-tail learning

In real-world scenarios, datasets often exhibit imbalanced class distributions, leading to models biased towards classes with a larger number of samples. This issue becomes more pronounced as the imbalance increases. Long-tailed datasets is defined imbalanced datasets where a few classes (head) have many samples and many classes (tail) have few samples. Long-tailed datasets typically have extremely imbalanced label distributions, often following an exponential distribution. Addressing this problem has led to extensive research in long-tail learning, as detailed in Appendix A.

Logit Adjustment (LA) (Menon et al., 2021) is a prominent method for tackling the long-tail learning task. The rationale behind LA is to adjust the model to be Bayes-optimal for balanced test sets by subtracting the logarithm of the label probability from the logits. This is formulated as follows:

$$\arg\max_{y \in [C]} \mathbb{P}^{\text{bal}}(y|x) := \arg\max_{y \in [C]} \mathbb{P}(x|y) = \arg\max_{y \in [C]} \mathbb{P}(y|x)/\mathbb{P}(y)$$
$$= \arg\max_{y \in [C]} \mathcal{S}(W^\top f_\theta(x))_y / \pi_y$$
$$= \arg\max_{y \in [C]} w_y^\top f_\theta(x)_y - \log \pi_y,$$

where $\mathcal{S}(W^\top f_\theta(x))_y$ is the softmax output for label $y$ given the weight matrix $W = [w_1, w_2, \ldots, w_C]$, $C$ is the number of classes, $[C] := \{1, \ldots, C\}$, and $\pi \in [0, 1]^C$ is the estimate of the label distribution over $C$ classes. In the formulation above, $\mathcal{S}(W^\top f_\theta(x))_y$ represents the output of the biased model trained on an imbalanced dataset. To achieve a Bayes-optimal classification model for a balanced distribution during training, the logits are adjusted by adding the logarithm of the label probability. The LA loss is defined as:

$$\mathcal{L}_{\text{LA}} = \mathbb{E}_{(x,y) \sim \mathcal{D}_{\text{ID}}} \Big[ \log[\sum_{c \in [C]} \exp (w_c^\top f_\theta(x) + \tau \log \pi_c)] - (w_y^\top f_\theta(x) + \tau \log \pi_y) \Big], \tag{2}$$

where $\tau > 0$ is a hyperparameter, which we set to 1.

Table 1: The existing OOD detection method (OE) and long-tail learning method (LA) aim to achieve better OOD detection and classification, respectively, by enforcing desired confidence gaps between ID vs. OOD or head vs. tail classes. Simply combining OE and LA (LA+OE) leads to trade-offs between OOD detection performance (FPR95) and classification accuracy, especially for the tail classes (Few). Our method (RNA) achieves superior OOD detection and classification performance by decoupling the two problems.

| Method | Softmax Confidence | | | FPR95 ($\downarrow$) | | Accuracy ($\uparrow$) | |
|---|---|---|---|---|---|---|---|
| | Avg. | Few | OOD | Avg. | Few | Avg. | Few |
| CE | 93.51 | 90.16 | 87.03 | 61.49 | 80.17 | 72.65 | 51.32 |
| CE+OE | 65.04 | 46.64 | 10.91 | 15.74 | 25.64 | 74.85 | 57.60 |
| LA | 92.75 | 89.54 | 81.65 | 52.64 | 66.04 | 77.30 | 64.18 |
| LA+OE | 64.01 | 46.63 | 10.94 | 17.92 | 28.45 | 75.32 | 57.93 |
| PASCL | 65.70 | 48.84 | 11.00 | 13.40 | 18.41 | 76.48 | 62.84 |
| RNA (ours) | 85.69 | 77.76 | 15.36 | **10.94** | **15.32** | **78.73** | **66.90** |

## 4  Motivation: trade-offs between OOD detection and long-tailed recognition

We first examine the OOD detection in long-tailed recognition to understand the main challenges. Specifically, we observe trade-offs between the two problems, when the state-of-the-art OOD detection method, Outlier Exposure (OE) (Hendrycks et al., 2019a), is simply combined with the widely used long-tail learning method, Logit Adjustment (LA) (Menon et al., 2021). OE exposes auxiliary OOD samples during training to discourage confident predictions on rare OOD samples by enforcing a uniform logit for OOD samples, while LA encourages confident predictions on rare tail-class ID samples by applying a label frequency-dependent offset to each logit during training.

In Table 1, we compare the performance of models trained with the cross-entropy (CE) loss, CE+OE loss (OE), LA, LA+OE, and PASCL (Wang et al., 2022b), which is a recently developed method for OOD detection in long-tailed recognition by using the contrastive learning (CL) idea, combined with LA+OE. We report the results of ResNet18 trained on CIFAR10-LT (Cui et al., 2019), when the performance is evaluated for an OOD test set, SVHN (Netzer et al., 2011). FPR95 is an OOD detection score, indicating the false positive rate (FPR) for ID data at a threshold achieving the true positive rate (TPR) of 95%. FPR95 can be computed for each class separately, and we report the FPR95 scores averaged over all classes (Avg.) and the bottom 33% of classes (Few) (in terms of the number of samples in the training set), and similarly for the classification accuracies.

Comparing the results for CE vs. CE+OE and for LA vs. LA+OE, we can observe that OE increases the gap between the softmax confidence of ID and OOD data, which makes it easier to distinguish ID data from OOD data by the confidence score, resulting in the lower FPR95. However, compared to the model trained with LA, the model trained with LA+OE achieves a lower classification accuracy, reduced by 1.98% on average and 6.25% for the 'Few' class group.

One can view LA+OE as a method that trades off the OOD detection performance and ID classification accuracy, especially for tail classes, since it achieves worse FPR95 but better classification accuracy compared to the OOD-tailored method, CE+OE, while it achieves better FPR95 but worse accuracy than the LTR-tailored method, LA. PASCL, which uses contrastive learning to push the representations of the tail-class ID samples and the OOD auxiliary samples, improves both the FPR95 and the classification accuracy for the 'Few' class group, each compared to those of CE+OE and LA, respectively. However, it still achieves a slightly lower average classification accuracy (-0.82%) than the model trained only for LTR by LA loss.

The main question is then whether such trade-offs between OOD detection and long-tailed recognition are fundamental, or whether we can achieve the two different goals simultaneously, without particularly compromising each other. All the existing methods reviewed in Table 1 aim to achieve better OOD detection, long-tailed recognition, or both by enforcing the desired logit distributions for ID vs. OOD data or head vs. tail classes, respectively. However, we observe the limitations of such approaches in achieving the two possibly conflicting goals simultaneously through the logit space, even with the help of contrastive learning.

Based on this observation, we propose to utilize a new dimension for OOD detection in the long-tail scenario, *the norm of representation vectors*, to decouple the OOD detection problem and the long-tailed recognition problem, and to achieve the two different goals without perturbing each other. As shown in Table 1, our method, Representation Norm Amplification (RNA), achieves the best FPR95 and classification accuracy for both the average case and the 'Few' class group, compared to other methods, including even those tailored to only one of the two goals, such as CE+OE or LA. In particular, our method achieves a desirable gap between the confidence scores of ID (Avg.) vs. OOD data (85.69 vs. 15.36%), while maintaining a relatively high confidence score of 77.76%, for the 'Few' ID classes.

# 5 Representation Norm Amplification (RNA)

In Section 5.1, we discuss the limitations of previous approaches in the way of exposing OOD samples during training by analyzing the loss gradients, and in Section 5.2, we present our method.

## 5.1 Previous ways of exposing auxiliary OOD data during training

Let $\mathcal{D}_{\text{ID}}$ and $\mathcal{D}_{\text{OOD}}$ denote an ID training dataset and an auxiliary OOD dataset, respectively. The state-of-the-art OOD detection methods expose the auxiliary OOD data during training to minimize the combination of a classification loss $\mathcal{L}_{\text{ID}}$ for ID data and an extra loss $\mathcal{L}_{\text{OOD}}$ for OOD data, i.e., $\mathcal{L} = \mathcal{L}_{\text{ID}} + \lambda\mathcal{L}_{\text{OOD}}$ for some tunable parameter $\lambda > 0$. As an example, one can choose the regular softmax cross-entropy loss as the ID loss, $\mathcal{L}_{\text{ID}} = \mathbb{E}_{(x,y)\sim\mathcal{D}_{\text{ID}}}\left[\log\left[\sum_{c\in[C]}\exp(w_c^\top f_\theta(x))\right] - \left(w_y^\top f_\theta(x)\right)\right]$, where $f_\theta(x) \in \mathbb{R}^D$ is the representation vector of the input data $x$ and $w_c \in \mathbb{R}^D$ is the classifier weight (i.e., the last-layer weight) for the label $c \in [C]$. For long-tailed recognition, the LA method (Menon et al., 2021) uses the logit-adjusted loss to improve the classification accuracy of ID data, i.e., $\mathcal{L}_{\text{ID}} = \mathcal{L}_{\text{LA}}$ in Equation 2. On the other hand, the loss for OOD training data is often designed to control the logit distribution or the softmax confidence of the OOD samples. For example, the OE method minimizes the cross entropy between the uniform distribution over $C$ classes and the softmax of the OOD sample, i.e., $\mathcal{L}_{\text{OOD}} = \mathbb{E}_{x'\sim\mathcal{D}_{\text{OOD}}}\left[\log\left[\sum_{c\in[C]}\exp(w_c^\top f_\theta(x'))\right] - \sum_{c\in[C]}\frac{1}{C}\left(w_c^\top f_\theta(x')\right)\right]$.

**OE perturbs tail classification: gradient analysis** We explain why training with OOD data to minimize the loss of the form $\mathcal{L} = \mathcal{L}_{\text{ID}} + \lambda\mathcal{L}_{\text{OOD}}$ can lead to a degradation in classification accuracy, particularly for tail classes. Let $\mathcal{T} = \{(x_i, y_i), x_i'\}_{i=1}^B$ be a training batch sampled from the sets of ID training samples $\{(x_i, y_i)\}$ and unlabeled OOD samples $\{x_i'\}$, respectively. For simpleness of analysis, we assume a fixed feature extractor $f_\theta(\cdot)$ and examine how ID/OOD samples in the batch affect the updates of the classifier weights $\{w_c\}_{c=1}^C$. Note that the gradient of the loss $\mathcal{L} = \mathcal{L}_{\text{ID}} + \lambda\mathcal{L}_{\text{OOD}}$ on the batch $\mathcal{T}$ with respect to the classifier weight $w_c$ is

$$\frac{\partial\mathcal{L}}{\partial w_c} = \frac{1}{B}\sum_{i=1}^B f_\theta(x_i)\left(\mathcal{S}(W^\top f_\theta(x_i))_c - y_i^c\right) + \frac{\lambda}{B}\sum_{i=1}^B f_\theta(x_i')\left(\mathcal{S}(W^\top f_\theta(x_i'))_c - 1/C\right), \tag{3}$$

where $y_i^c = 1$ if $y_i = c$ and 0 otherwise. For $\mathcal{L}_{\text{ID}} = \mathcal{L}_{\text{LA}}$, $\mathcal{S}(W^\top f_\theta(x_i))_c$ in the first term is replaced by the softmax output for the logits adjusted by the label-dependent offsets as in Equation 2.

From Equation 3, the gradient is a weighted sum of the representations of ID data $\{f_\theta(x_i)\}_{i=1}^B$ and OOD data $\{f_\theta(x_i')\}_{i=1}^B$, where the weights depend on the gap between the softmax output $\mathcal{S}(W^\top f_\theta(x_i))_c$ and the desired label, $y_i^c$ for ID data and $1/C$ for OOD data, respectively. Thus, the gradient balances the effects of ID samples, tuned to minimize the classification error, and those of OOD samples, tuned to control their confidence levels. For a tail class $t$, however, the proportion of ID samples with the label $y_i = t$ is relatively small. As a result, during training, the gradient for the classifier weight $w_t$ of the tail class is dominated by the representations of OOD samples rather than those of the tail-class ID samples, particularly at the early stage of training when the label distribution for OOD samples deviates significantly from a uniform distribution. In Figure 2a, we plot the log-ratios $(\log(\|\nabla_{w_c}\mathcal{L}_{\text{ID}}\|_1/\|\nabla_{w_c}\mathcal{L}_{\text{OOD}}\|_1))$ of the gradient norms between ID and OOD samples with respect to the classifier weights of bottom 3 (tail) classes vs. top 3 (head) classes. Classifier weights for tail classes are more heavily influenced by OOD samples than those for head classes, especially in the early stages of training, resulting in a greater degradation of classification accuracy,

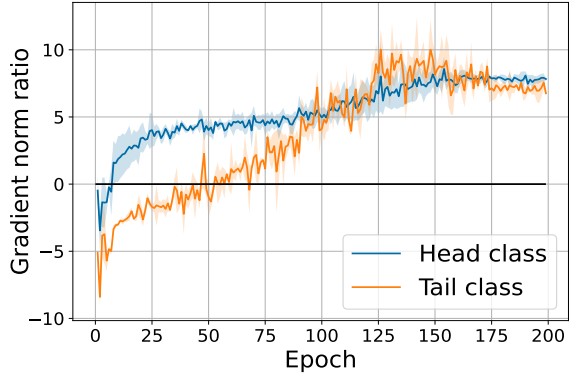
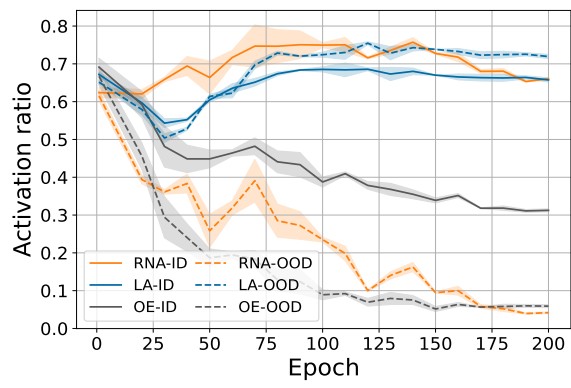

(a) Log-ratio of gradient norms of head/tail class data

(b) Activation ratio at the last ReLU layer

Figure 2: (a) The gradient ratio of ID classification loss to OOD detection loss with respect to the classifier weight of LA+OE model trained on CIFAR10-LT, i.e. $\log(\|\nabla_{w_c}\mathcal{L}_{\mathrm{ID}}\|_1/\|\nabla_{w_c}\mathcal{L}_{\mathrm{OOD}}\|_1)$. In particular, $\mathcal{L}_{\mathrm{LA}}$ and $\mathcal{L}_{\mathrm{OE}}$ are used as $\mathcal{L}_{\mathrm{ID}}$ and $\mathcal{L}_{\mathrm{OOD}}$, respectively. Note that the log-ratio for tail classes is less than zero at the early stage of training, indicating that the gradient update is dominated by OOD data rather than ID data. (b) The activation ratio of ID and OOD representations at the last ReLU layer in the models trained by RNA, LA, and OE. CIFAR10 and SVHN are used as ID and OOD sets, respectively.

as observed in Table 1. The question is then how to effectively expose OOD data during training to learn the representations to distinguish ID and OOD data, while not degrading the classification.

## 5.2 Proposed OOD scoring and training method with auxiliary OOD data

**Representation Norm (RN) score** We present a simple yet effective approach to expose OOD data during training to create a discrepancy in the representation between ID and OOD data. The main strategy is to utilize the norm of representations (i.e., feature vectors). Note that the representation vector $f_\theta(x)$ can be decomposed into its norm $\|f_\theta(x)\|_2$ and direction $\hat{f}_\theta(x) = f_\theta(x)/\|f_\theta(x)\|_2$. Simply scaling the norm $\|f_\theta(x)\|_2$ of the representation without changing its direction does not change the class prediction, since for any scaling factor $s > 1$, we have $\arg\max_{c \in [C]} \mathcal{S}(W^\top s f_\theta(x))_c = \arg\max_{c \in [C]} \mathcal{S}(W^\top f_\theta(x))_c$. We thus leverage the norm of representations to generate a discrepancy between the representations of ID vs. OOD data, while not enforcing the desired logit distribution for OOD data. This approach, termed as *Representation Norm (RN) score*, is formulated as

$$S_{\mathrm{RN}}(x) = -\|f_\theta(x)\|_2, \tag{4}$$

for a test sample $x$. Following the convention for OOD detection, samples with larger scores are detected as OOD data, while those with smaller scores are considered as ID data.

**Representation Norm Amplification (RNA)** We propose a training method to obtain desired representations for ID vs. OOD data. Our goal in the representation learning is twofold: (i) to make the norms of the representations for ID data relatively larger than those of OOD data, and (ii) to learn the representations that achieve high classification accuracy even when trained on long-tailed datasets. To achieve the first goal, we propose Representation Norm Amplification (RNA) loss $\mathcal{L}_{\mathrm{RNA}}$:

$$\mathcal{L}_{\mathrm{RNA}} = \mathbb{E}_{x \sim \mathcal{D}_{\mathrm{ID}}} \left[ -\log(1 + \|h_\phi(f_\theta(x))\|_2) \right], \tag{5}$$

where $h_\phi$ is a 2-layer MLP projection function. This loss is designed to increase the norm of the representations for ID data, employing a logarithmic transformation on the L2-norm of the representation. The gradient of the RNA loss with respect to the projected representation is expressed as:

$$\frac{\partial \hat{\mathcal{L}}_{\mathrm{RNA}}}{\partial h_i} = -\frac{1}{(1 + \|h_i\|_2)} \frac{h_i}{\|h_i\|_2}, \tag{6}$$

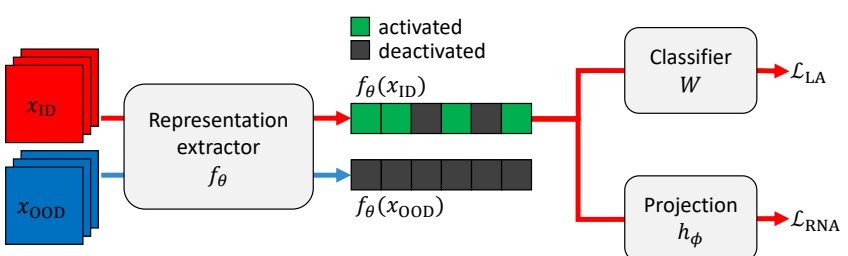
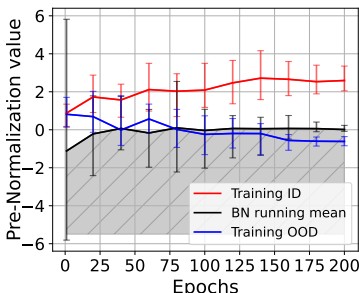

(a) Overview of the proposed method, Representation Norm Amplification (RNA)   (b) Values before the last BN layer

Figure 3: (a) During training, RNA uses both ID and auxiliary OOD data. The network parameters are updated to minimize the classification loss $\mathcal{L}_{\text{LA}}$ (Equation 2) of the ID samples, regularized by their representation norms through the RNA loss $\mathcal{L}_{\text{RNA}}$ (Equation 5). The OOD data only indirectly contribute to the updating of the model parameters, as being used to update the running statistics of the BN layers. (b) This illustration represents the latent values of ID (red) and OOD (blue) samples before the last BN layer of the RNA-trained model. The error bars denote the maximum and minimum values among the latent vectors averaged over ID and OOD data, respectively. Through the training, the running mean of the BN layer (black) converges between the ID and OOD values. After passing through the last BN and ReLU layers (before the classifier), the shaded region beneath the BN running mean is deactivated. RNA effectively generates a noticeable gap in the activation ratio at the last ReLU layer and the representation norm between ID vs. OOD data, which serves as the basis for our OOD score.

where $\hat{\mathcal{L}}_{\text{RNA}}$ represents the empirical estimation of RNA loss, $\hat{\mathcal{L}}_{\text{RNA}} := -\frac{1}{B} \sum_{i=1}^{B} \log(1 + \|h_\phi(f_\theta(x_i))\|_2)$, with a batch size $B$, and $h_i$ denotes the projected representation of a sample $x_i$, $h_i := h_\phi(f_\theta(x_i))$. The norm of this gradient is computed as:

$$\left\| \frac{\partial \hat{\mathcal{L}}_{\text{RNA}}}{\partial h_i} \right\|_2 = \frac{1}{(1 + \|h_i\|_2)}. \tag{7}$$

The gradient norms are inversely proportional to representation norms. Consequently, during the training process, the magnitude of updates to model parameters decreases as ID representation norms increase. This behavior promotes convergence of the training procedure. Further details on training dynamics can be found in Appendix E.9. We then combine this loss with the logit adjustment (LA) loss $\mathcal{L}_{\text{LA}}$ (Menon et al., 2021), tailored for long-tail learning, and define our final loss function as below:

$$\mathcal{L} = \mathcal{L}_{\text{LA}} + \lambda \mathcal{L}_{\text{RNA}}, \tag{8}$$

where $\lambda > 0$ is a hyperparameter balancing the two objectives. We set $\lambda = 0.5$ for all the experiments in Section 6.

It is important to note that we only use ID data to optimize the loss function (Equation 8) by gradient descent. OOD training data do not provide gradients for model parameter updates, but instead are simply fed into the model as shown in Figure 3a. We use OOD data to regularize the model by updating the running statistics of the BN layers from both ID and OOD data. OOD samples contribute to updating only the running mean and variance of BN layers, causing changes to the latent vectors of ID data. However, the OOD representations do not directly participate in the loss or gradient for the classifier weights. Thus, OOD data only indirectly influences parameter updates. In the rest of this section, we will explain why forwarding OOD data is sufficient to regularize the model to produce relatively smaller representation norms of OOD data compared to those of ID data, when the total loss is regularized by $\mathcal{L}_{\text{RNA}}$.

**Effect of auxiliary OOD data: regularizing the activation ratio**   In common deep neural network architectures, BN layers are often followed by ReLU layers (Nair & Hinton, 2010). BN layers normalize inputs by estimating the mean and variance of all the given data. Consequently, the coordinates of input vectors with values below the running mean of the BN layer are deactivated as they pass through the next ReLU

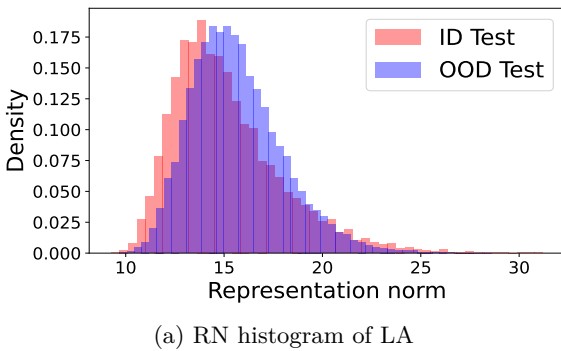

(a) RN histogram of LA

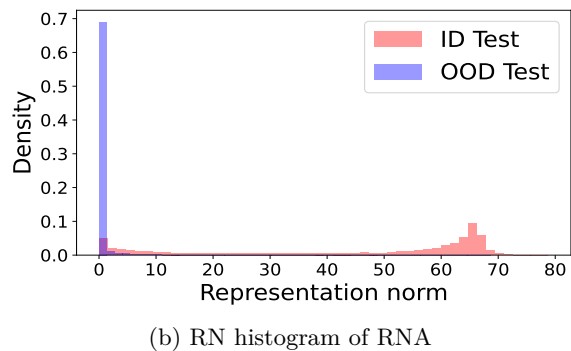

(b) RN histogram of RNA

Figure 4: (a) The histogram of representation norms of ID/OOD test data on LA-trained models on CIFAR10-LT. The red bars represent the density of representation norms of ID data and the blue bars represent that of OOD data (SVHN). (b) The histogram of representation norms of ID/OOD test data on RNA-trained models on CIFAR10-LT. The evident gap in the distributions of representation norms of ID and OOD data enables effective OOD detection using representation norms.

layer (Figure 3b). In Figure 2b, we compare the activation ratio (the fraction of activated coordinates) at the last ReLU layer (prior to the last linear layer) for ID data (CIFAR-10), and OOD data (SVHN), using models trained by RNA, LA, and OE. RNA exhibits distinct activation ratio patterns, clearly separating ID and OOD data. RNA activates the majority of the coordinates of representation vectors of ID data (66.08% on CIFAR10) while deactivating those of OOD data (4.17% on SVHN). This indicates that BN layer regularization using both ID and OOD data, along with enlarging the ID representation norms through $\mathcal{L}_{\text{RNA}}$, results in a substantial discrepancy in the activation ratio at the last ReLU layer. Moreover, since the activation ratio corresponds to the fraction of non-zero entries in the representation vector, a notable gap in the activation ratio signifies a difference in representation norms between ID and OOD data, facilitating reliable OOD detection by the RN score (Equation 4). Figure 4a and 4b present histograms of representation norms of ID vs. OOD data when trained by LA and RNA, respectively. The evident gap between ID and OOD test sets is shown only for the RNA-trained models, enabling effective OOD detection with representation norms. Additional RN histograms for RNA-trained models on other pairs of ID and OOD test sets are available in Appendix E.10. The training dynamics of RNA, including loss dynamics of LA loss (Equation 2) and RNA loss (Equation 5), as well as representation norm dynamics of head and tail data in ID training/test datasets, are provided in Appendix E.9.

## 6 Experimental results

### 6.1 Experimental setup

**Datasets and training setup** For ID datasets, we use CIFAR10/100 (Krizhevsky, 2009) and ImageNet-1k (Deng et al., 2009). The long-tailed training sets, CIFAR10/100-LT, are built by downsampling CIFAR10/100 (Cui et al., 2019), making the imbalance ratio, $\max_c N(c)/\min_c N(c)$, equal to 100, where $N(c)$ is the number of samples in class $c$. The results for various imbalance ratios, including balanced scenario, are reported in Table 5. We use 300K random images (Hendrycks et al., 2019a) as an auxiliary OOD training set for CIFAR10/100. For ImageNet, we use ImageNet-LT (Liu et al., 2019) as the long-tailed training set and ImageNet-Extra (Wang et al., 2022b) as the auxiliary OOD training set.

For experiments on CIFAR10/100, we use the semantically coherent out-of-distribution (SC-OOD) benchmark datasets (Yang et al., 2021) as our OOD test sets. For CIFAR10 (respectively, CIFAR100), we use Textures (Cimpoi et al., 2014), SVHN (Netzer et al., 2011), CIFAR100 (respectively, CIFAR10), Tiny ImageNet (Le & Yang, 2015), LSUN (Yu et al., 2015), and Places365 (Zhou et al., 2018) from the SC-OOD benchmark. For ImageNet, we use ImageNet-1k-OOD (Wang et al., 2022b), as our OOD test set. All the OOD training sets and OOD test sets are disjoint. Further details on datasets can be found in Appendix D.3. We train ResNet18 (He et al., 2015) on CIFAR10/100-LT datasets for 200 epochs with a batch size of 128, and ResNet50 (He

Table 2: OOD detection performance (AUC, AUPR, and FPR95) and ID classification performance (ACC, Many, Medium, and Few) (%) on CIFAR10/100-LT and ImageNet-LT. Means and standard deviations are reported based on six runs. **Bold** indicates the best performance, and underline indicates the second best performance. FT represents that the models are fine-tuned with ID samples for LTR tasks.

| Method | AUC (↑) | AUPR (↑) | FPR95 (↓) | ACC (↑) | Many (↑) | Medium (↑) | Few (↑) |
|---|---|---|---|---|---|---|---|
| | | | ID Dataset: CIFAR10-LT | | | | |
| MSP | 72.68±1.04 | 70.48±0.89 | 65.67±1.82 | 72.65±0.19 | **94.65±0.27** | 72.19±0.49 | 51.32±0.74 |
| OECC | 88.08±0.19 | 88.65±0.09 | 48.48±0.88 | 74.48±0.19 | 94.29±0.29 | 72.69±0.68 | 56.79±0.68 |
| EnergyOE | 88.78±0.48 | 88.91±0.61 | 44.48±0.78 | 76.25±0.61 | 93.94±0.51 | 74.37±0.34 | 61.01±1.87 |
| OE | 89.69±0.51 | 86.47±1.41 | 33.81±0.51 | 74.85±0.57 | 93.90±0.35 | 73.30±1.12 | 57.60±1.78 |
| PASCL | 91.00±0.26 | 89.35±0.52 | 33.51±0.87 | 75.86±0.63 | 92.50±0.22 | 72.84±0.94 | 63.67±1.76 |
| BEL | 92.49±0.06 | 91.83±0.06 | 30.88±0.21 | 76.30±0.17 | 91.58±0.14 | 74.54±0.29 | 63.38±0.50 |
| RNA (ours) | **92.90±0.27** | **92.14±0.36** | **29.18±0.90** | 78.73±0.23 | 93.72±0.17 | **76.40 ±0.85** | 66.90±1.06 |
| PASCL-FT | 88.03±0.79 | 84.66±2.03 | 40.43±1.22 | 76.48±0.22 | 92.93±0.71 | 74.20±1.11 | 62.84±1.01 |
| BEL-FT | 90.80±0.18 | 90.57±0.22 | 36.79±0.65 | **81.50±0.19** | 89.63±0.40 | 76.33±0.58 | **80.26±0.40** |
| | | | ID Dataset: CIFAR100-LT | | | | |
| MSP | 61.39±0.40 | 57.86±0.30 | 82.17±0.38 | 40.61±0.20 | **71.02±0.67** | 38.59±0.45 | 11.06±0.28 |
| OECC | 69.64±0.47 | 66.55±0.40 | 76.74±0.58 | 41.52±0.37 | 69.88±0.31 | 39.43±0.78 | 13.97±0.45 |
| EnergyOE | 69.14±0.53 | 66.67±0.21 | 80.04±1.13 | 39.17±0.34 | 68.53±0.52 | 39.54±0.47 | 8.03±0.82 |
| OE | 73.37±0.22 | 67.26±0.41 | 67.83±0.82 | 39.83±0.32 | 68.28±0.63 | 37.11±0.45 | 12.89±0.49 |
| PASCL | 73.14±0.37 | 66.77±0.50 | 67.36±0.46 | 40.01±0.31 | 68.97±0.48 | 37.18±0.38 | 12.35±0.46 |
| BEL | **77.66±0.05** | **73.01±0.06** | **61.18±0.14** | 40.87±0.04 | 61.85±0.14 | **47.98±0.16** | 12.56±0.18 |
| RNA (ours) | 75.06±0.49 | 70.71±0.56 | 66.90±1.05 | 44.80±0.25 | 68.35±0.68 | 44.97±0.36 | 19.84±0.48 |
| PASCL-FT | 73.59±0.58 | 67.69±0.62 | 67.66±0.49 | 43.44±0.34 | 65.83±0.38 | 40.76±0.60 | 22.70±0.94 |
| BEL-FT | 75.07±0.31 | 70.71±0.34 | 67.96±0.32 | **45.53±0.14** | 61.30±0.38 | 44.11±0.35 | **31.22±0.44** |
| | | | ID Dataset: ImageNet-LT | | | | |
| MSP | 54.22±0.66 | 51.85±0.54 | 89.67±0.44 | 41.69±1.45 | 59.01±1.90 | 35.54±2.49 | 14.29±5.15 |
| OECC | 62.82±0.28 | 63.89±0.33 | 87.83±0.20 | 41.03±0.23 | 59.36±0.36 | 34.32±0.25 | 12.78±0.34 |
| EnergyOE | 63.38±0.20 | 64.51±0.23 | 88.34±0.19 | 38.47±1.00 | 58.82±0.99 | 30.61±1.17 | 8.56±0.52 |
| OE | 67.12±0.35 | 69.20±0.34 | 87.65±0.20 | 40.87±0.32 | **60.20±0.26** | 34.07±0.42 | 10.11±0.65 |
| PASCL | 68.17±0.24 | 70.26±0.31 | 87.62±0.32 | 40.94±0.94 | 60.00±0.92 | 34.24±1.04 | 10.59±0.73 |
| BEL | 64.90±0.14 | 65.92±0.18 | 87.91±0.36 | 40.54±0.88 | 60.27±0.91 | 33.40±0.98 | 9.84±0.56 |
| RNA (ours) | **75.55±0.23** | **74.60±0.27** | 78.16±0.81 | **47.81±0.31** | 58.59±0.38 | **44.58±0.38** | **28.67±0.31** |
| PASCL-FT | 61.83±0.33 | 62.86±0.59 | 88.95±0.26 | 44.97±1.01 | 55.98±1.08 | 42.29±1.06 | 23.26±0.84 |
| BEL-FT | 58.26±0.28 | 57.29±0.30 | 90.33±0.36 | 45.08±1.00 | 55.11±1.01 | 42.15±0.96 | 27.01±1.25 |

et al., 2015) on ImageNet-LT for 100 epochs with a batch size of 256. We set $\lambda = 0.5$ in (8). More details on the implementation are available in Appendix D.1.

**Evaluation metrics** To evaluate OOD detection, we use three metrics –the area under the receiver operating characteristic curve (AUC), the area under the precision-recall curve (AUPR), and the false positive rate of ID samples at the threshold of true positive rate of 95% (FPR95). For ID classification, we measure the total accuracy (ACC) and also the accuracy averaged for a different subset of classes categorized as 'Many', 'Medium', and 'Few' in terms of the number of training samples. All the reported numbers are the average of six runs from different random seeds.

## 6.2 Results

**Results on CIFAR10/100** The results on CIFAR10/100-LT are summarized in Table 2, where the OOD detection performances are averaged over six different OOD test sets. We compare against several baselines, including MSP (Hendrycks & Gimpel, 2017), OECC (Papadopoulos et al., 2021), EngeryOE (Liu et al., 2020), OE (Hendrycks et al., 2019a), and two recently published methods tailored for LT-OOD tasks, PASCL (Wang

Table 3: Performance metrics (%) for OOD detection and ID classification, along with the representation norm and the activation ratio at the last ReLU layer, comparing variants of RNA loss on CIFAR10-LT. Results for our original proposed method (RNA) are highlighted in gray.

| | Training loss | | | | | | Representation norm | | Activation ratio | |
| $\mathcal{L}_{\text{LA}}$ | $\mathcal{L}_{\text{OOD-att.}}$ | $\mathcal{L}_{\text{ID-amp.}}$ | AUC($\uparrow$) | FPR95($\downarrow$) | ACC($\uparrow$) | ACC-Few($\uparrow$) | ID test | OOD test | ID test | OOD test |
|---|---|---|---|---|---|---|---|---|---|---|
| ✓ | ✓ | | 82.68 | 55.05 | 77.82 | 65.49 | 28.84 | 14.55 | 0.55 | 0.55 |
| ✓ | ✓ | ✓ | 84.03 | 41.59 | 72.92 | **69.62** | 43.65 | 13.14 | 0.74 | 0.18 |
| ✓ | | ✓ | **92.90** | **29.18** | **78.73** | 66.90 | 43.79 | 0.67 | 0.66 | 0.02 |

et al., 2022b) and BEL (Choi et al., 2023). We categorize the models based on whether they incorporate separate fine-tuning for LTR after OOD detection training. Certain baselines perform OOD detection and ID classification with two separate models, where one is trained for OOD detection (PASCL and BEL) while the other is fine-tuned for LTR (PASCL-FT and BEL-FT). Notably, the RNA-trained models perform OOD detection and ID classification with a single model. RNA outperforms the baseline methods for OOD detection on CIFAR10-LT and ranks second on CIFAR100-LT. In particular, on CIFAR10-LT, RNA improves AUC, AUPR, and FPR95 by 0.41%, 0.31% and 1.70%, respectively, compared to the state-of-the-art method, BEL. RNA also shows improvements of 2.43% and 3.23% over the accuracy for the toal and 'Few' class groups, respectively, compared to the second best models among those without fine-tuning. When trained on CIFAR100-LT, RNA achieves the second-best AUC, AUPR, and FPR95 with values of 75.06%, 70.71%, and 66.90%, respectively. Moreover, RNA exhibits the highest overall classification accuracy of 44.80%, except for the BEL-FT. The results of OOD detection performance on each OOD test set are reported in Appendix E.1.

**Results on ImageNet** Table 2 also shows the results on ImageNet-LT, which is a large scale dataset. Our method significantly improves the OOD detection performance by 7.38%, 4.34%, and 9.46% for AUC, AUPR, and FPR95, respectively, compared to the second best model, PASCL. Furthermore, RNA outperforms BEL-FT in terms of the classification accuracy, increased by 2.73% and 1.66% for total and 'Few' cases, respectively. It is noteworthy that the performance improvement of our method on ImageNet-LT is greater than those on CIFAR10/100-LT, indicating that our method demonstrates effective performance on large scale datasets, which is considered as a more challenging case for OOD detection. Furthermore, pairs of models, such as PASCL and PASCL-FT, and BEL and BEL-FT, still exhibit trade-offs between OOD detection and ID classification. PASCL-FT improves in ACC and 'Few' by 4.03% and 12.67%, respectively, compared to PASCL. However, PASCL-FT underperforms PASCL in OOD detection by 6.34%, 7.40%, and 1.33% in AUC, AUPR, and FPR95, respectively. Similarly, BEL-FT outperforms BEL in ACC and 'Few' by 4.54% and 17.17%, respectively, but lags in AUC, AUPR, and FPR95 by 6.64%, 8.63%, and 2.42%, respectively. In contrast, RNA achieves the state-of-the-art performance in both OOD detection and ID classification with a single model, overcoming these trade-offs.

## 6.3 Additional experiments

We perform a series of ablation study to investigate our method from multiple perspectives. Our evaluations involve variants of RNA training loss, diverse combinations of OOD training and scoring methods, different imbalance ratios, RNA-trained models with different configurations of auxiliary OOD sets, and calibration performance. Further ablation studies are available in Appendix E, where we include the results about the robustness to $\lambda$ in Equation 8 (§E.2), trade-offs in models trained by LA+OE (§E.3), the size of auxiliary OOD sets (§E.4), the training batch ratios of ID and OOD samples (§E.5), alternative auxiliary OOD sets (§E.6), the structure of the projection function $h_\phi$ (§E.7), and OOD fine-tuning task (§E.8).

### 6.3.1 Variants of RNA training loss

We conduct an ablation study on variants of the RNA loss. Our proposed RNA loss ($\mathcal{L}_{\text{ID-amp.}}$) amplifies ID representation norms during training. We examine two alternative variants: a loss to attenuate OOD representation norms, denoted to as $\mathcal{L}_{\text{OOD-att.}}$, and a combination of both losses. The OOD-attenuation loss is defined as $\mathcal{L}_{\text{OOD-att.}} = \mathbb{E}_{x \sim \mathcal{D}_{\text{OOD}}}[\log(1 + \|h_\phi(f_\theta(x))\|)]$, which is the negative counterpart of the RNA

Table 4: AUC (%) of various combinations of training methods and OOD scores on CIFAR10-LT, CIFAR100-LT and ImageNet-LT. The gray regions indicate the results for our proposed methods. MSP and Energy are confidence-based OOD scoring methods, while RN is evaluated in the representation space.

| Training Method | OOD Scoring Method | | | | | | | | |
|---|---|---|---|---|---|---|---|---|---|
| | CIFAR10-LT | | | CIFAR100-LT | | | ImageNet-LT | | |
| | MSP | Energy | RN (ours) | MSP | Energy | RN (ours) | MSP | Energy | RN (ours) |
| CE | 72.68 | 72.77 | 70.98 | 61.39 | 61.55 | 55.67 | 54.22 | 54.07 | 56.73 |
| OE | 89.69 | 89.66 | 86.63 | 73.37 | 72.98 | 71.81 | 67.12 | 67.77 | 77.99 |
| PASCL | 91.00 | 91.09 | 81.18 | 73.14 | 72.56 | 70.68 | 68.17 | 68.70 | 77.98 |
| BEL | 87.96 | 92.49 | 92.30 | 70.86 | 77.66 | 76.43 | 59.53 | 64.90 | 75.37 |
| RNA (ours) | 91.73 | 91.96 | 92.90 | 73.93 | 74.00 | 75.06 | 61.27 | 61.50 | 75.55 |

loss in Equation 5. As shown in Table 3, these variants are not as effective as the original RNA loss on CIFAR10-LT. As shown in Table 3, the combination of attenuation loss and amplification loss widens the gap in representation norm and activation ratio compared to the case of using only the attenuation loss. However, the amplification loss alone creates a more substantial gap than the combination loss. In particular, the combination loss succeeds in enlarging the ID norms, but whenever attenuation loss is used (either alone or with amplification loss), the resulting OOD norm is not as low as the case of using only the amplification loss.

The rationale behind this result lies in the optimization process. When solely employing the amplification loss (RNA loss), the optimizer amplifies the norm $\|h_\phi(f_\theta(x))\|$ for ID data, necessarily by promoting high activation ratios for ID data while reducing the activation ratio for OOD data (as low as 2% in our experiment), due to the BN regularization using both ID and OOD data. However, applying the attenuation loss (either alone or in combination form) does not necessarily suppress activations for OOD samples in the feature coordinates to minimize the loss. Instead, the loss can be minimized by mapping the OOD features into its null space. Consequently, the resulting activation ratios and the representation norms for OOD samples are not as low as the case of using only the amplification loss.

### 6.3.2 Scoring methods

We evaluate the effects of our training method (Equation 8) and scoring method (Equation 4) separately. Table 4 presents the mean AUC over six OOD test sets for models trained with different objectives: CE, OE, PASCL, BEL and RNA, and evaluated with various OOD scoring methods: MSP, Energy, and RN. On CIFAR10-LT, the RNA-trained model achieves the highest AUC for MSP and RN scoring, and the second-best for Energy scoring. On CIFAR100-LT, RNA ranks the first with MSP and second with Energy and RN. This can be attributed to the fact that RNA training indirectly increases the confidence of ID data by amplifying representation norms. Consequently, RNA maintains competitive AUC performance even with MSP or Energy scoring, which rely on the confidence level. Additionally, our RN scoring method performs comparably to MSP or Energy scoring for the models trained by OE, PASCL, and BEL. While these methods regularize the confidence of auxiliary OOD samples, they lead to a significant gap in representation norms between ID vs. OOD data (as shown in Figure 2b), which allows our RN score to detect OOD samples.

The experiments with ImageNet-LT show different trends. The RNA-trained model exhibits a lower AUC with confidence-based scores (MSP and Energy) compared to the models trained using OE or PASCL. However, when RN scoring is used, all the training methods (OE, PASCL, BEL and RNA) achieve significant improvements of 10.22%, 9.28%, 10.47%, and 14.05% over Energy scoring, respectively. This highlights the superiority of the representation-based OOD scoring method (RN), especially for a large scale dataset.

### 6.3.3 Imbalance ratio

In Table 5, we present the AUC and ACC metrics for the models trained on CIFAR10/100-LT with different imbalance ratios. Higher ratios indicate greater imbalance, posing a more challenging training scenario. A ratio of 1 represents a balanced dataset, not a long-tailed one. Our proposed method consistently outperforms PASCL in both AUC and ACC across the imbalance ratios of 1, 10, 50, 100, and 1000, including a balanced

Table 5: AUC (%) and classification accuracy (%) of models trained on CIFAR10-LT and CIFAR100-LT with various imbalance ratio $\rho$, including balanced training sets ($\rho = 1$).

| Method | AUC / ACC | | | | | | | | |
|---|---|---|---|---|---|---|---|---|---|
| $\rho$ | 1 | 10 | 50 | 100 | 1000 | 1 | 10 | 50 | 100 |
| OE | 96.78/93.55 | 94.32/87.47 | 91.35/79.62 | 89.69/74.85 | 82.98/57.21 | 84.56/71.32 | 78.57/55.78 | 74.72/44.13 | 73.37/39.83 |
| PASCL | 96.67/93.67 | 94.68/87.62 | 92.25/80.15 | 91.00/76.48 | 86.62/59.43 | **84.64**/71.89 | **78.59**/57.89 | 75.02/47.16 | 73.14/43.44 |
| RNA (ours) | **97.07/94.24** | **95.86/89.28** | **93.87/82.29** | **92.90/78.73** | **88.93/60.72** | 81.79/**74.10** | 78.19/**60.17** | **75.67/48.81** | **74.33/44.39** |

Table 6: OOD detection and ID clsasification performance (%) on CIFAR10-LT with different usage settings of auxiliary set.

| Method | Auxiliary set | | AUC | ACC |
|---|---|---|---|---|
| | Aux. OOD | Aug. ID | | |
| MSP | | | 72.68 | 72.65 |
| RNA | | | 75.57 | 79.32 |
| | | ✓ | 83.57 | 78.13 |
| | ✓ | | 92.90 | 78.73 |

Table 7: Calibration performance measured by Expected Calibration Error (ECE) (%) on CIFAR10, CIFAR100, and ImageNet.

| Method | CIFAR10 | CIFAR100 | ImageNet |
|---|---|---|---|
| CE | 20.76±0.27 | 32.06±0.25 | 21.55±1.86 |
| OE | 17.56±0.61 | **16.53±0.33** | 16.83±0.41 |
| PASCL | 18.78±0.63 | 17.17±0.25 | 16.37±0.48 |
| RNA (ours) | **12.56±0.49** | 17.67±0.41 | **12.65±0.42** |

training set. On the other hand, when trained on CIFAR100-LT, RNA improves ACC across all the imbalance ratios, but it exhibits relatively lower AUC when the imbalance ratio is 1 or 10. These results shows the superiority of our model for various imbalance ratios of the training set, especially for high imbalance ratios.

### 6.3.4 Use of auxiliary OOD data

In our ablation study, we investigate the impact of incorporating auxiliary OOD data in our method. The results in Table 6 reveal that employing the RNA loss without any auxiliary data leads to a modest AUC improvement of merely 2.89% over the baseline MSP approach. In contrast, the combination of RNA loss with BN regularization, including auxiliary OOD samples, yields a substantial gain of 20.22%. This clearly highlights the significant role of BN regularization using auxiliary OOD data, coupled with the RNA loss.

We also explore the potential for eliminating the need for auxiliary samples. We conduct an experiment wherein the auxiliary OOD set is replaced with augmented images derived from the original training samples. We employ a random cropping augmentation strategy spanning 5% to 25% of the original image size. This strategy yields a notable AUC gain of 10.89% over the MSP baseline and an 8.00% gain compared to not utilizing any auxiliary samples but applying RNA loss. While not matching the performance gains of our original method with the auxiliary OOD set, this suggests the potential for replacing auxiliary OOD samples with augmented training samples in applying our method.

### 6.3.5 Calibration

We assess the calibration performance of our method on CIFAR10/100-LT and ImageNet-LT. To measure calibration, we utilize Expected Calibration Error (ECE), defined as $\text{ECE} := \sum_{m=1}^{M} \frac{|B_m|}{n} |\text{ACC}(B_m) - \text{Conf}(B_m)|$, where $M$ and $n$ are the number of bins and test data, respectively, $B_m := \{x : (m-1)/M < \text{Conf}(x) \leq m/M\}$ represents the confidence bin, and $\text{ACC}(B_m)$ and $\text{Conf}(B_m)$ denote average accuracy and confidence within the bin $B_m$, respectively. In Table 7, we report the measured ECE values for models trained with CE, OE, PASCL, and RNA. The models are trained on the long-tailed datasets, and the ECE values are evaluated on the balanced test sets. RNA achieves the highest ECE values for CIFAR10 and ImageNet. This result demonstrates that our method exhibits strong calibration performance for ID data.

## 7 Discussion

We propose a simple yet effective method for OOD detection in long-tail learning, Representation Norm Amplification (RNA). Our method regularizes the representation norm of only ID data, while updating

the BN statistics using both ID and OOD data, which effectively generates a noticeable discrepancy in the representation norm between ID and OOD data. The effectiveness of our method is demonstrated with experimental results on various OOD test sets and long-tailed training sets. Broader impacts, limitations and further discussion of our work are available in Appendix B.

## 8    Acknowledgement

This work was supported by the National Research Foundation of Korea (NRF) grant funded by the Korea government (MSIT) (No. RS-2024-00408003 and No. 2021R1C1C11008539).

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

# A    Detailed review on related works

In this section, we provide more detailed review on some of the related works, discussed in Sec. 2.

**OOD detection**   OOD scores are typically evaluated based on output confidence (Bendale & Boult, 2016; Hendrycks & Gimpel, 2017; Liang et al., 2018; Liu et al., 2020), feature space properties (Lee et al., 2018b; Dong et al., 2021; Sun et al., 2021; Song et al., 2022; Sun & Li, 2022; Sun et al., 2022; Vaze et al., 2022; Wang et al., 2022a; Ahn et al., 2023; Djurisic et al., 2023; Yu et al., 2023; Zhu et al., 2022b), or gradients (Huang et al., 2021). Many methods have been developed to encourage the separability of ID and OOD data by these scores, for example by exposing auxiliary OOD data during training and regularizing their confidence levels (Malinin & Gales, 2018; Hein et al., 2019; Liu et al., 2020; Mohseni et al., 2020; Papadopoulos et al., 2021; Yang et al., 2021; Katz-Samuels et al., 2022; Ming et al., 2022), or by promoting the ID-OOD separability in the embedding space (Dhamija et al., 2018; Lee et al., 2018a; Hendrycks et al., 2019b; Choi & Chung, 2020; Hsu et al., 2020; Tack et al., 2020; Sehwag et al., 2021; Wang et al., 2021; Ming et al., 2023). However, these methods often suffer from performance degradation in a long-tail learning setup (Wang et al., 2022b).

**Long-tail learning**   Long-tail learning addresses the classification problem when trained on imbalanced or long-tailed class distributions. When trained on imbalanced datasets, the model is often biased towards the majority classes, generating underconfident predictions on the minority or tail classes. There have been many methods to fix this problem by over- or under-sampling (Kubat et al., 1997; Chawla et al., 2002; Pouyanfar et al., 2018; Kang et al., 2021), adjusting margins or decision thresholds (Cao et al., 2019; Tan et al., 2020; Menon et al., 2021), or modifying the loss functions (Menon et al., 2021; Cui et al., 2021; Li et al., 2022; Zhu et al., 2022a). In particular, some methods attempt to increase the value of the logit for the tail classes by the post-hoc normalization of the weight of the classifier (the last linear layer) (Kang et al., 2020), or by applying a label frequency-dependent offset to each logit during training (Menon et al., 2021). While these approaches have shown empirical effectiveness and statistical robustness in the context of long-tail learning, they often conflict with the principle of out-of-distribution (OOD) detection methods, which regulate the confidence of models on rare samples to prevent overconfident predictions (Hendrycks et al., 2019a; Papadopoulos et al., 2021).

**OOD detection and controlling overconfidence**   There exist OOD detection methods that do not use auxiliary outlier data during training, but effectively mitigate the overconfidence issue of the models. For example, LogitNorm (Wei et al., 2022) characterizes that training with cross-entropy loss causes logit norms to keep increasing as the training progresses, and this often leads to overconfident predictions of the models on both in-distribution and OOD test data. Therefore, the LogitNorm method normalizes the logit of every input data during training to prevent the logit norm from increasing too much. Some recently published works, Rectified Activations (ReAct) (Sun et al., 2021) and Directed Sparisification (DICE) (Sun & Li, 2022), on the other hand, focus on the fact that on a model trained only on in-distribution data, OOD data still activate a non-negligible fraction of units in the penultimate layer, where the mean activation is biased towards having sharp positive values and there are "noisy units" with high variances of contribution to the class output. Thus, these methods use post-hoc activation truncation (Sun et al., 2021) or weight sparsification (Sun & Li, 2022), to avoid overconfident predictions on OOD data.

**OOD detection with abstention class**   OOD detection methods can be categorized into score-based methods and methods incorporating abstention classes. Score-based methods employ specific OOD scoring metrics to measure the OOD-ness of test samples such as MSP (Hendrycks & Gimpel, 2017), Energy (Liu et al., 2020), and our proposed method, RN. RNA and all the baseline methods including MSP (Hendrycks & Gimpel, 2017), OECC (Papadopoulos et al., 2021), EnergyOE (Liu et al., 2020), OE (Hendrycks et al., 2019a), PASCL (Wang et al., 2022b), and BEL (Choi et al., 2023) belong to the score-based OOD detection category. On the other hand, OOD detection methods with an abstention class use networks with additional class for OOD data, referred to as abstention classes. Consequently, the network outputs $(C+1)$-dimensional vectors, where $C$ is the number of classes. During training, these methods often employ classification loss to categorize auxiliary OOD samples into the abstention class. These methods faced challenges that OOD representations tended to collapse, being treated as the same class, despite dissimilarities in image semantics

among OOD samples. However, recent studies (Mohseni et al., 2020; Chen et al., 2021) strive to overcome these shortcomings and achieve improved OOD detection performance in balanced scenarios.

## B    Broader impacts and limitations

Mitigating OOD uncertainty is crucial, particularly in safety-critical applications such as medical diagnosis and autonomous driving. In these domains, the ability to effectively identify and handle OOD samples is essential to ensure the reliability and safety of the systems. By addressing the challenges associated with OOD detection, we can significantly enhance the trustworthiness and applicability of deep learning models in these critical domains.

Our proposed method specifically focuses on tackling the challenging scenarios where reliable OOD detection needs to be performed on models trained with long-tailed distributions, which is a common occurrence in practical settings but has received limited attention in previous research. By introducing our method, we enhance the safety and reliability of deploying deep models, even when trained on imbalanced datasets that have not been extensively preprocessed for class balance. This enables the application of deep learning models in real-world scenarios, where long-tailed distributions are prevalent, without compromising their OOD detection capabilities.

Our proposed method does have some limitations. Firstly, it relies on the availability of an auxiliary OOD dataset for effective performance. We address this issue by investigating the possibility of replacing the auxiliary OOD dataset by augmented training dataset as reported in Table 6.

Additionally, as shown in Table 8, our method demonstrates suboptimal performance on the near OOD test set, where the OOD data is semantically similar to the training data. This limitation arises because the representation distributions of the near OOD test data are closer to the in-distribution training data rather than the auxiliary OOD data, resulting in some test OOD samples having large representation norms. This vulnerability to near OOD data can be alleviated by employing confidence-based OOD scoring methods. However, it entails a trade-off in OOD detection performance for far OOD data, as discussed in Sec. E.1. To address this limitation, future research should focus on developing training methods that can enhance the OOD detection performance specifically for near OOD test sets in long-tail learning scenarios. By overcoming this challenge, we can further strengthen the robustness and practical applicability of OOD detection methods in more broad real-world settings.

## C    Pseudocode

The training scheme of Representation Norm Amplification (RNA) is shown in Algorithm 1, and evaluation scheme of Representation Norm (RN) is shown in Algorithm 2.

## D    Details on the experimental setup

### D.1    Implementation details

We train ResNet18 (He et al., 2015) on CIFAR10/100-LT datasets for 200 epochs with a batch size of 128. We optimize the model parameters with Adam optimizer (Kingma & Ba, 2014) with an initial learning rate of 0.001 and we decay the learning rate with the cosine learning scheduler (Loshchilov & Hutter, 2017). The weight decay parameter is set to 0.0005.

We train ResNet50 (He et al., 2015) on ImageNet-LT for 100 epochs with a batch size of 256. We optimize the model parameters with SGD optimizer with a momentum value of 0.9 and an initial learning rate of 0.1, and we decay the learning rate with the cosine learning scheduler. The weight decay parameter is set to 0.0005.

We set the balancing hyperparameter $\lambda = 0.5$ as the default value. We do not use the learnable affine parameters $\beta$ and $\gamma$ of all BN layers in the model, since they are trained to amplify both ID and OOD data by converging to large values. During training, the batch consists of an equal number of ID and OOD data samples. For example, a batch size of 256 means that it contains 256 ID samples and 256 OOD samples.

---

**Algorithm 1:** RNA: Representation Norm Amplification

---

**Input:** In-distribution training set $\mathcal{D}_{\text{ID}}$, auxiliary OOD training set $\mathcal{D}_{\text{OOD}}$, batch size $b$, number of classes $C$, representation extractor $f_\theta$, linear classifier $W = [w_1, w_2, \dots, w_C]$, projection function $h_\phi$, learning rate $\eta$

**1 for** $epoch = 1, 2, \dots,$ **do**

**2**      **for** $iter = 1, 2, \dots,$ **do**

         // Sample a batch of ID and OOD samples.

**3**          $\mathcal{B}_{\text{ID}} = \{(x_i, y_i)\}_{i=1}^{b} \sim \mathcal{D}_{\text{ID}}$ & $\mathcal{B}_{\text{OOD}} = \{x_i\}_{i=b+1}^{2b} \sim \mathcal{D}_{\text{OOD}}$

**4**          $\mathcal{B} = \mathcal{B}_{\text{ID}} \cup \mathcal{B}_{\text{OOD}}$

         // Feed forward the batch into $f_\theta$.

**5**          $[z_1, z_2, \dots, z_{2b}] = f_\theta([x_1, x_2, \dots, x_{2b}])$

         // Calculate the loss function with only ID representations.

**6**          $\mathcal{L}_{\text{LA}} = \sum_{i=1}^{b} \left[ \log[\sum_{c \in [C]} \exp{(w_c^\top z_i + \log \pi_c)}] - (w_{y_i}^\top z_i + \log \pi_{y_i}) \right]$

**7**          $\mathcal{L}_{\text{RNA}} = \sum_{i=1}^{b} \left[ -\log(1 + \|h_\phi(z_i)\|) \right]$

**8**          $\mathcal{L} = \mathcal{L}_{\text{LA}} + \lambda \mathcal{L}_{\text{RNA}}$

         // Update the model parameters.

**9**          $\theta \longleftarrow \theta - \eta \nabla_\theta \mathcal{L}$

**10**         $W \longleftarrow W - \eta \nabla_W \mathcal{L}$

**11**         $\phi \longleftarrow \phi - \eta \nabla_\phi \mathcal{L}$

---

**Algorithm 2:** RN: Representation Norm Score

---

**Input:** Test set $\mathcal{D}_{\text{test}}$, representation extractor $f_\theta$, linear classifier $W = [w_1, w_2, \dots, w_C]$

**1 for** $x_{\text{test}} \in \mathcal{D}_{\text{test}}$ **do**

     // Feed forward a test sample into $f_\theta$.

**2**      $z_{\text{test}} = f_\theta(x_{\text{test}})$

     // Calculate the RN score for OOD detection.

**3**      $S_{\text{RN}}(x_{\text{test}}) = -\|z_{\text{test}}\|_2$

     // Obtain the softmax prediction for ID classification.

**4**      $p_{\text{test}} = \arg\max \mathcal{S}(W^\top z_{\text{test}})$

---

For baseline methods, we implement the methods following the training setting reported in the original papers.

### D.2 Computational resource and time

We run the experiments on NVIDIA A6000 GPUs. For CIFAR10/100-LT datasets, we use a single GPU for each experimental run, and the entire training process takes about an hour. For ImageNet-LT, we use 7 GPUs for each experimental run, and the entire training process takes about 4 and a half hours.

### D.3 Datasets

For the CIFAR benchmark, we employ 300K random images as an auxiliary OOD training set following Hendrycks et al. (2019a). 300K random images is a subset of 80M Tiny Images dataset (Torralba et al., 2008). The selection process for the 300K random images ensures that the image classes are disjoint with those in the CIFAR10 and CIFAR100 datasets.

For the experiments on ImageNet-1k, we use ImageNet-Extra (Wang et al., 2022b) as an auxiliary OOD training set, and ImageNet-1k-OOD, published in Wang et al. (2022b), as our OOD test set. ImageNet-Extra is created by sampling 517,711 data points from 500 randomly chosen classes in ImageNet-22k, where the classes are disjoint from the classes in ImageNet-1k. ImageNet-1k-OOD contains 50,000 OOD test images

evenly sampled from 1,000 randomly selected classes of ImageNet-22k, where the 1,000 classes do not overlap with those of ImageNet-1k and ImageNet-Extra.

### D.4 Baseline methods

In this section, we summarize the main baseline methods that we compare to our method in the experiments.

- **Outlier Exposure (OE) (Hendrycks et al., 2019a):** OE is a training method specifically designed for out-of-distribution (OOD) detection, with the objective of amplifying the discrepancy in softmax confidence between in-distribution (ID) and OOD data. It leverages an auxiliary OOD dataset effectively during the training process to enhance the model's ability to distinguish between ID and OOD samples. The OE method involves constructing training batches that consist of both ID training samples and auxiliary OOD samples. By combining these samples, the method aims to minimize a composite loss function that includes both the classification loss for ID samples and an additional loss term for OOD samples. This additional loss term is defined as the cross-entropy between the uniform distribution and the softmax probabilities of the OOD samples.

- **Energy OE (Liu et al., 2020):** EnergyOE (Liu et al., 2020) is a variant of the OE method that offers an alternative approach to controlling the confidence levels of OOD samples. Instead of regularizing the cross-entropy loss of auxiliary OOD samples, EnergyOE maximizes the free energy of OOD samples. By maximizing the free energy, EnergyOE aims to increase the uncertainty and reduce the confidence associated with OOD samples, facilitating their detection and differentiation from in-distribution samples.

- **Outlier Exposure with Confidence Control (OECC) (Papadopoulos et al., 2021):** OECC is another variant of the OE method, which controls the total variation distance between the network output for OOD training samples and the uniform distribution, regularized by Euclidean distance between the training accuracy and the average confidence on the model's prediction on the training set.

- **Partial and Asymmetric Supervised Contrastive Learning (PASCL) (Wang et al., 2022b):** PASCL (Wang et al., 2022b), which is a recently developed method for OOD detection in long-tailed recognition, uses the ideas from supervised contrastive learning (Khosla et al., 2020), combined with Logit Adjustment (LA) (Menon et al., 2021) and outlier exposure (OE) (Hendrycks et al., 2019a), to achieve both high classification accuracy and reliable OOD detection performance. In particular, this method applies the partial and asymmetric contrastive learning that pushes the representations of tail-class in-distribution samples and those of OOD training samples, while pulling only tail-class in-distribution samples of the same class. In addition, in the second stage, the Batch Normalization (BN) (Ioffe & Szegedy, 2015) layers and the last linear layer are fine-tuned using only ID training data in order for the running mean and standard deviation of BN layers to be re-fit to the ID data distribution only, since the BN layers are fitted to the union of ID and auxiliary OOD data distribution before the second stage.

- **Balanced Energy Learning (BEL) (Choi et al., 2023):** BEL (Choi et al., 2023) is an enhanced algorithm derived from Energy OE (Liu et al., 2020). Energy OE (Liu et al., 2020) trains the model to output smaller energy for ID samples and larger energy for auxiliary OOD samples. While both methods share the scheme of energy regularization for OOD detection, BEL (Choi et al., 2023) weighs auxiliary OOD samples individually based on the pseudo-label of each sample. Initially, the prior label distribution is estimated on the auxiliary OOD set by the model undergoing training. Subsequently, in the loss term and energy margin, auxiliary OOD samples estimated as head class are more emphasized. This approach effectively achieves superior OOD detection performance, but it still requires auxiliary fine-tuning stage for achieving high classification accuracy in long-tailed scenarios, akin to PASCL (Wang et al., 2022b).

# E  Additional experimental results

## E.1  OOD detection performance on six different OOD test sets

Table 8: AUC (%) for six different OOD test sets on models trained with CIFAR10/100-LT.

| Method | OOD test set | | | | | | Average |
| | Texture | SVHN | CIFAR | Tiny ImageNet | LSUN | Places365 | |
|---|---|---|---|---|---|---|---|
| | ID Dataset: CIFAR10-LT | | | | | | |
| MSP | 72.91±1.64 | 71.98±2.82 | 70.94±0.58 | 72.86±0.66 | 74.61±0.89 | 72.80±0.72 | 72.68±1.04 |
| OECC | 91.26±0.59 | 95.86±1.00 | 80.29±0.24 | 83.69±0.09 | 90.38±0.45 | 87.02±0.21 | 88.08±0.19 |
| EnergyOE | 91.73±0.56 | 92.39±2.01 | 81.70±0.51 | 84.57±0.53 | 92.26±0.26 | 90.02±0.17 | 88.78±0.48 |
| OE | 91.66±0.72 | 95.02±1.13 | 83.43±0.32 | 86.10±0.33 | 91.81±0.62 | 90.15±0.44 | 89.69±0.51 |
| PASCL | 93.35±0.72 | 96.65±0.54 | 84.44±0.31 | 87.34±0.30 | 92.84±0.29 | 91.37±0.19 | 91.00±0.26 |
| BEL | 95.56±0.02 | **97.44±0.27** | 85.14±0.05 | 88.76±0.04 | 94.66±0.08 | 93.39±0.08 | 92.49±0.06 |
| RNA (ours) | **95.97±0.33** | 97.36±0.65 | **85.47±0.30** | **89.33±0.25** | **95.48±0.17** | **93.76±0.12** | **92.90±0.27** |
| | ID Dataset: CIFAR100-LT | | | | | | |
| MSP | 55.68±0.47 | 63.45±2.32 | 60.42±0.29 | 62.60±0.28 | 62.48±0.34 | 63.70±0.18 | 61.39±0.40 |
| OECC | 69.74±1.32 | 70.41±1.26 | 60.21±0.51 | 67.59±0.32 | 75.79±0.61 | 74.11±0.40 | 69.64±0.47 |
| EnergyOE | 69.96±0.91 | 73.46±3.93 | 61.25±0.43 | 66.29±0.23 | 72.21±0.46 | 71.66±0.20 | 69.14±0.53 |
| OE | 76.56±0.79 | 79.31±1.72 | **62.44±0.58** | 68.51±0.34 | 77.41±0.34 | 75.99±0.34 | 73.37±0.22 |
| PASCL | 76.15±0.60 | 79.23±1.39 | 62.26±0.17 | 68.37±0.25 | 77.02±0.31 | 75.79±0.24 | 73.14±0.37 |
| BEL | **81.99±0.09** | **88.32±0.18** | 59.26±0.21 | **71.33±0.07** | **83.92±0.10** | **81.11±0.09** | **77.66±0.05** |
| RNA (ours) | 78.38±0.56 | 85.85±2.64 | 58.04±0.53 | 68.20±0.27 | 80.99±0.37 | 78.89±0.35 | 75.06±0.49 |

Table 9: AUPR (%) for six different OOD test sets on models trained with CIFAR10/100-LT.

| Method | OOD test set | | | | | | Average |
| | Texture | SVHN | CIFAR | Tiny ImageNet | LSUN | Places365 | |
|---|---|---|---|---|---|---|---|
| | ID Dataset: CIFAR10-LT | | | | | | |
| MSP | 54.24±1.84 | 82.52±1.56 | 66.11±0.65 | 62.76±0.71 | 70.20±0.96 | 87.05±0.42 | 70.48±0.89 |
| OECC | 87.56±0.75 | 98.28±0.45 | 80.11±0.22 | 79.83±0.14 | 90.91±0.46 | 95.22±0.09 | 88.65±0.09 |
| EnergyOE | 86.71±1.11 | 95.75±1.19 | 81.73±0.62 | 80.85±0.72 | 92.22±0.34 | 96.22±0.12 | 88.91±0.61 |
| OE | 79.52±3.63 | 96.74±1.26 | 80.18±1.04 | 78.15±0.99 | 89.03±1.70 | 95.23±0.54 | 86.47±1.41 |
| PASCL | 85.74±1.99 | 98.14±0.63 | 83.05±0.50 | 81.82±0.70 | 91.11±0.58 | 96.24±0.09 | 89.35±0.52 |
| BEL | **92.12±0.09** | **98.66±0.13** | 84.85±0.05 | 84.78±0.10 | 93.32±0.12 | 97.23±0.04 | 91.83±0.06 |
| RNA (ours) | 92.08±0.73 | 98.61±0.33 | **85.04±0.41** | **85.10±0.48** | **94.65±0.26** | **97.40±0.07** | **92.14±0.36** |
| | ID Dataset: CIFAR100-LT | | | | | | |
| MSP | 38.86±0.31 | 77.80±1.56 | **57.82±0.33** | 46.50±0.26 | 46.42±0.54 | 79.78±0.13 | 57.86±0.30 |
| OECC | 57.26±1.59 | 82.67±0.99 | 56.15±0.34 | 52.76±0.39 | 63.62±0.86 | 86.82±0.23 | 66.55±0.40 |
| EnergyOE | 58.34±1.38 | 86.22±2.22 | 57.73±0.55 | 52.02±0.42 | 59.85±0.65 | 85.89±0.16 | 66.67±0.21 |
| OE | 58.70±2.18 | 87.54±0.96 | 57.33±0.53 | 52.26±0.59 | 61.25±0.59 | 86.53±0.26 | 67.26±0.41 |
| PASCL | 57.50±1.85 | 87.09±0.75 | 57.02±0.15 | 51.85±0.44 | 60.88±0.65 | 86.31±0.18 | 66.77±0.50 |
| BEL | **72.92±0.14** | **92.69±0.16** | 54.86±0.15 | **56.41±0.11** | **71.24±0.13** | **89.94±0.04** | **73.01±0.06** |
| RNA (ours) | 67.56±1.24 | 91.97±2.01 | 53.93±0.55 | 53.68±0.41 | 68.15±0.50 | 88.97±0.27 | 70.71±0.56 |

Table 8, 9 and 10 present the AUC, AUPR and FPR95 metrics, respectively, for the models trained using RNA and other baseline approaches across the six OOD test sets. These results collectively demonstrate the OOD detection performance. Specifically, in Table 8, RNA achieves the best results for all the six OOD test sets when the model is trained on CIFAR10-LT. For CIFAR100-LT, RNA achieves the best results on four OOD test sets as well as on average, except for CIFAR10 and Tiny ImageNet, which are semantically more similar to the training set (CIFAR100-LT) than the other OOD test sets. In Table 9, our proposed method exhibits the highest AUPR values across all the OOD test sets when trained on CIFAR10-LT. Similarly, for

Table 10: FPR95 (%) for six different OOD test sets on models trained with CIFAR10/100-LT.

| Method | OOD test set | | | | | | Average |
|---|---|---|---|---|---|---|---|
| | Texture | SVHN | CIFAR | Tiny ImageNet | LSUN | Places365 | |
| ID Dataset: CIFAR10-LT | | | | | | | |
| MSP | 67.27±3.88 | 61.49±5.34 | 70.52±1.56 | 65.70±0.97 | 62.11±1.33 | 66.93±1.13 | 65.67±1.82 |
| OECC | 43.70±1.45 | 24.62±3.89 | 63.02±0.79 | 56.39±0.71 | 47.77±1.26 | 55.36±0.63 | 48.48±0.88 |
| EnergyOE | 37.89±1.19 | 29.84±3.62 | 63.42±1.81 | 55.82±1.41 | 36.37±1.40 | 43.54±0.60 | 44.48±0.78 |
| OE | 23.57±0.67 | 15.74±2.37 | 56.30±0.46 | 46.26±0.93 | 27.81±0.60 | 33.19±0.59 | 33.81±0.51 |
| PASCL | 22.91±1.74 | 13.40±1.72 | 57.20±0.85 | 47.26±0.21 | 26.91±0.94 | 33.35±0.87 | 33.51±0.87 |
| BEL | 22.22±0.39 | **10.88±1.03** | 57.94±0.19 | 43.04±0.13 | 23.13±0.39 | 28.10±0.42 | 30.88±0.21 |
| RNA (ours) | **18.02±1.38** | 10.94±2.91 | **55.82±1.36** | **42.11±0.27** | **20.50±0.57** | **27.72±0.42** | **29.18±0.90** |
| ID Dataset: CIFAR100-LT | | | | | | | |
| MSP | 89.44±0.88 | 75.22±2.55 | 86.00±0.53 | 81.59±0.23 | 80.85±0.47 | 79.93±0.32 | 82.17±0.38 |
| OECC | 80.37±2.18 | 70.63±2.75 | 85.32±0.57 | 78.42±0.41 | 72.41±0.59 | 73.28±0.24 | 76.74±0.58 |
| EnergyOE | 84.40±1.04 | 75.12±5.22 | 81.45±0.48 | 81.15±0.64 | 79.64±1.20 | 78.50±0.76 | 80.04±1.13 |
| OE | 67.08±2.13 | 55.36±3.62 | 79.85±0.71 | 76.17±0.83 | 63.24±0.31 | 65.31±0.40 | 67.83±0.82 |
| PASCL | 65.50±1.64 | 53.57±3.21 | **79.81±0.35** | 76.37±0.38 | 64.09±0.88 | 64.83±0.41 | 67.36±0.46 |
| BEL | **64.16±0.23** | **34.06±0.16** | 85.27±0.53 | **74.54±0.16** | **51.54±0.13** | **57.52±0.32** | **61.18±0.14** |
| RNA (ours) | 69.60±1.75 | 45.47±4.97 | 83.43±0.53 | 79.32±0.33 | 60.36±1.10 | 63.23±0.47 | 66.90±1.05 |

Table 11: The AUROC performance (%) of the model trained with RNA using different OOD scoring method. The OOD test sets for near OOD task are CIFAR10 and Tiny ImageNet, while those for far OOD task are Texture, SVHN, LSUN, and Places365.

| Training method | Scoring method | Near OOD | Far OOD | Average |
|---|---|---|---|---|
| OE | MSP | **65.48** | 77.32 | 73.37 |
| RNA | MSP | 64.81 | 78.49 | 73.93 |
| RNA | RN | 62.63 | **80.19** | **74.33** |

all but the CIFAR10 test set, RNA achieves the highest AUPR values when trained on CIFAR100-LT. In Table 10 our proposed method shows the best FPR95 values across all but the CIFAR100 test sets when trained on CIFAR10-LT. On the other hand, while RNA achieves the lowest FPR95 values for SVHN, LSUN, and Places365 when trained on CIFAR100-LT, it is outperformed by PASCL in terms of the average FPR95 value across the OOD test sets.

Near OOD detection poses a significant challenge in the field of OOD detection. In these scenarios, the OOD test sets closely resemble the ID datasets. RNA method exhibits suboptimal performance for near OOD detection as shown in Table 8, 9 and 10. This is particularly noticeable when using CIFAR100-LT as the ID training set and CIFAR10 as the OOD test set. In contrast, other methods like OE with MSP score exhibit relatively robust performance in such settings.

We attribute this observed trend to the inherent characteristics of the CIFAR10 and CIFAR100 datasets. While these datasets consist of disjoint classes, certain classes share overlapping features. For instance, presenting data from the "dog" class of CIFAR10 OOD test set to a CIFAR100-trained model may activate features associated with CIFAR100 classes like "fox", "raccoon", "skunk", "otter", and "beaver". Consequently, the logits corresponding to these multiple classes may simultaneously have high values, potentially leading to a low MSP value. However, the representation norm might be large due to these related features. For such an example, the near OOD sample may be detected by MSP but incorrectly categorized as an ID sample by the RN score. To support this hypothesis, Table 11 illustrates the OOD detection performance (AUC) for two OOD scoring methods, MSP and RN, when applied to the RNA-trained model on CIFAR100-LT. As anticipated, using MSP leads to an improvement in near OOD detection performance, but at the cost of worsening far OOD detection.

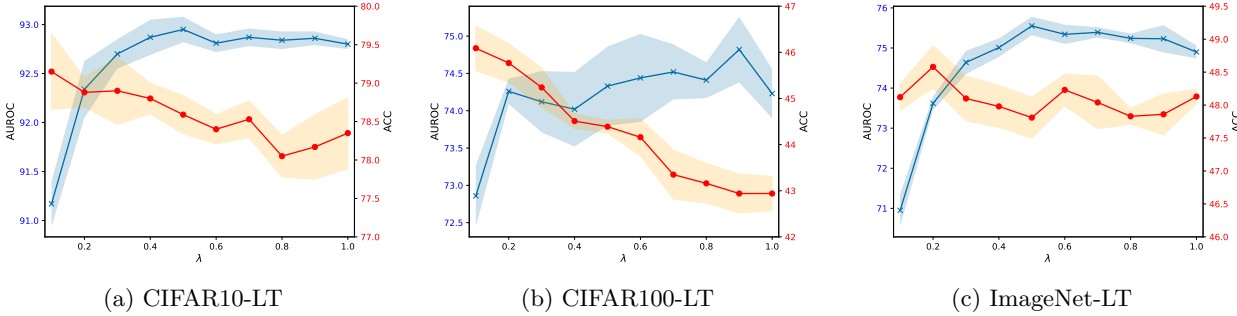

(a) CIFAR10-LT           (b) CIFAR100-LT           (c) ImageNet-LT

Figure 5: AUC and ACC (%) performance of RNA-trained model with varying the balancing hyperparameter $\lambda$. Blue (x-marked) lines indicate the AUC values and red (circle-marked) lines indicate the ACC values.

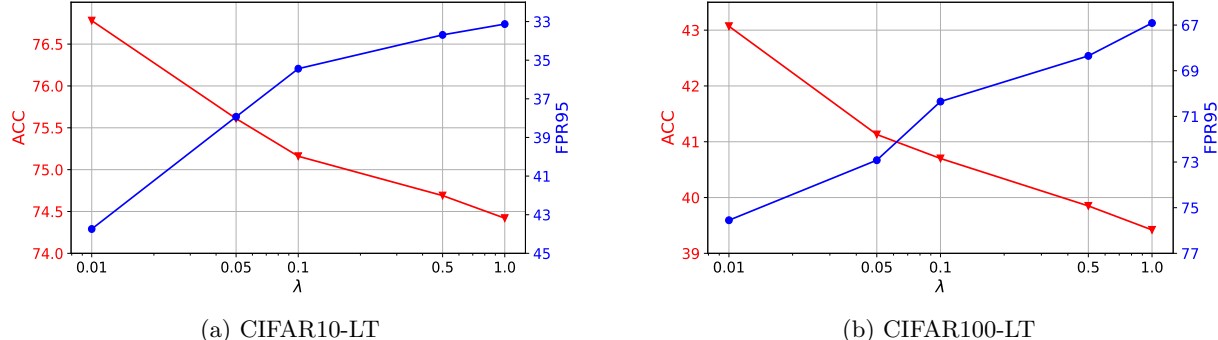

(a) CIFAR10-LT                           (b) CIFAR100-LT

Figure 6: The ID classification performance (ACC) and OOD detection performance (FPR95) of models trained by LA+OE. FPR95 is averaged over six OOD test sets as in Section 6.1. The models are trained with the training loss, $\mathcal{L}_{\mathrm{LA}} + \lambda\mathcal{L}_{\mathrm{OE}}$, where $\lambda$ denotes the balancing hyperparameter. The results illustrate the trade-offs inherent in combining long-tail learning methods (LA) and OOD detection methods (OE).

## E.2 Hyperparameter robustness

Our proposed training method uses the combination of the two loss terms, the classification loss and the RNA loss regularizing the representation norm of input data, as shown in Equation 8. The relative importance of the two loss terms can be tuned with the hyperparameter $\lambda$. Although all the results in the main paper are obtained with the fixed value of $\lambda = 0.5$, we extensively explore the performance results when models are trained with varying the hyperparameter $\lambda$ for CIFAR10-LT, CIFAR100-LT, and ImageNet-LT. In Figure 5, we plot the AUC and ACC performances for 10 values of $\lambda = 0.1, 0.2, \ldots, 1.0$. For CIFAR10/100-LT and ImageNet-LT, the AUC values (blue lines) are robust to the change in $\lambda$ when $\lambda$ is greater than 0.2, although they are much lower when $\lambda$ is small. On the other hand, the classification accuracy (red lines) decreases as $\lambda$ increases when trained on CIFAR10-LT and CIFAR100-LT, and this tendency is more dramatic when trained on CIFAR100-LT. For ImageNet-LT, the accuracy does not change significantly as $\lambda$ values change.

## E.3 Trade-offs in LA+OE-trained models

We explore the influence of LA and OE in LA+OE models by adjusting the balancing parameter $\lambda$ in the loss function, $\mathcal{L}_{\mathrm{LA}} + \lambda\mathcal{L}_{\mathrm{OE}}$. Figure 6 presents the results of LA+OE models trained on CIFAR10-LT and CIFAR100-LT with various $\lambda$ values of 0.01, 0.05, 0.1, 0.5, and 1.0. Across both datasets, LA+OE-trained models exhibit superior ID classification performance but inferior OOD detection performance with small $\lambda$, while showing superior OOD detection performance but inferior ID classification performance with large $\lambda$. These findings highlight the trade-offs between ID classification and OOD detection performance.

### E.4  Size of auxiliary OOD sets

Table 12: OOD detection and ID classification performance (%) across different number of auxiliary OOD samples on CIFAR10-LT.

| The number of ID samples | The number of OOD samples | AUC($\uparrow$) | FPR95($\downarrow$) | ACC($\uparrow$) | ACC-Few($\uparrow$) |
|---|---|---|---|---|---|
| | 300 | 67.75 | 74.26 | 45.82 | 39.47 |
| 12406 | 3k | 91.10 | 37.54 | 80.18 | 74.98 |
| | 30k | 92.90 | 29.46 | 79.01 | 68.32 |
| | 300k | 92.90 | 29.18 | 78.73 | 66.90 |

We present the performance results of RNA-trained models varying the size of auxiliary OOD training sets on CIFAR10-LT in Table 12. The OOD detection and ID classification performance remain comparable when the number of OOD samples is 3k, 30k, or 300k. However, when the number of OOD samples is reduced to 300, which is approximately 2.42% of the number of ID training samples, the AUC and ACC drop rapidly by 23.35% and 34.36%, respectively. In conclusion, RNA-trained models achieve comparable performance in both OOD detection and ID classification when the number of OOD samples is at least a quarter of the ID samples. However, the OOD samples are too few, the performance of the RNA-trained model degrades significantly.

### E.5  Training batch ratio of ID and OOD samples

Table 13: Performance metrics (%) for OOD detection and ID classification, along with the representation norm and the activation ratio at the last ReLU layer, comparing the training batch ratio on CIFAR10-LT.

| Batch ratio (ID:OOD) | AUC($\uparrow$) | FPR95($\downarrow$) | ACC($\uparrow$) | Representation norm | | Activation ratio | | $\frac{\mu_{\mathrm{ID}}-\mu_{\mathrm{BN}}}{\mu_{\mathrm{BN}}-\mu_{\mathrm{OOD}}}$ |
|---|---|---|---|---|---|---|---|---|
| | | | | ID test | OOD test | ID test | OOD test | |
| 128:32 | 91.87 | 35.37 | 79.00 | 36.55 | 2.56 | 54.55 | 6.10 | 0.13 |
| 128:64 | 92.62 | 31.50 | 79.36 | 38.28 | 1.89 | 56.31 | 3.46 | 0.23 |
| 128:128 | 92.90 | 29.18 | 78.73 | 43.79 | 0.67 | 62.64 | 2.08 | 0.37 |
| 128:256 | 93.04 | 27.89 | 79.02 | 55.08 | 0.70 | 75.34 | 1.43 | 0.80 |
| 128:512 | 92.95 | 27.64 | 79.19 | 64.12 | 0.45 | 72.26 | 1.12 | 0.90 |

In Table 13, we present experiments varying the OOD sample size. Our method demonstrates worse FPR95 performances with smaller OOD sample sizes.

Our approach detects OOD data based on representation norms, which are proportional to the gap between latent vectors before the last BN layer and the running mean of the last BN layer. So, we will check the gaps with varying the OOD sample size. Let $\mu_{\mathrm{ID}}$ and $\mu_{\mathrm{OOD}}$ denote the mean of the latent vectors before the last BN layer for ID and OOD samples, respectively, and $\mu_{\mathrm{BN}}$ denote the running mean of the last BN layer. After training with RNA, we observe that $\mu_{\mathrm{ID}} > \mu_{\mathrm{BN}} > \mu_{\mathrm{OOD}}$, as shown in Figure 3b, which implies that many ID representations are activated, while many OOD representations are deactivated after RNA training.

For superior OOD detection based on representation norms, ID representation norms should be relatively larger than OOD representation norms. Thus, $\frac{\mu_{\mathrm{ID}}-\mu_{\mathrm{BN}}}{\mu_{\mathrm{BN}}-\mu_{\mathrm{OOD}}}$ is proportional to the OOD detection performance. At different ratio of ID to OOD, different number of ID or OOD samples accounts for calculating $\mu_{\mathrm{BN}}$, since $\mu_{\mathrm{BN}}$ is the mean of all the ID and OOD samples in the batch. Therefore, as the OOD sample size decreases, $\mu_{\mathrm{BN}}$ gets closer to $\mu_{\mathrm{ID}}$ than $\mu_{\mathrm{OOD}}$, so the ratio $\frac{\mu_{\mathrm{ID}}-\mu_{\mathrm{BN}}}{\mu_{\mathrm{BN}}-\mu_{\mathrm{OOD}}}$ decreases, as shown in the rightmost column of Table 13. This decreased ratio $\frac{\mu_{\mathrm{ID}}-\mu_{\mathrm{BN}}}{\mu_{\mathrm{BN}}-\mu_{\mathrm{OOD}}}$ leads to smaller gap of representation norms between ID and OOD and declined OOD detection performance.

Table 14: OOD detection and ID classification performance (%) on CIFAR10-LT with ImageNet-RC (Chrabaszcz et al., 2017) as the auxiliary OOD training set.

| Method | AUC (↑) | AUPR (↑) | FPR95 (↓) | ACC (↑) | Many (↑) | Medium (↑) | Few (↑) |
|---|---|---|---|---|---|---|---|
| OE | 92.29 | **91.85** | 28.16 | 74.67 | **94.59** | 73.64 | 56.10 |
| PASCL | 92.05 | 91.70 | 29.72 | 73.58 | 94.43 | 72.76 | 53.88 |
| RNA (ours) | **92.53** | 91.78 | **23.61** | **78.67** | 94.13 | **75.44** | **67.67** |
| PASCL-FT | 90.66 | 89.44 | 33.88 | 77.78 | 93.53 | 73.88 | 67.12 |

## E.6 Alternative auxiliary OOD training set

We additionally conduct experiments employing ImageNet-RC as an auxiliary OOD dataset for CIFAR10-LT ID dataset. The results are presented in Table 14. Our results demonstrate the effectiveness of the RNA, which consistently outperforms both the OE and PASCL, in terms of OOD detection and ID classification. While the AUC and AUPR metrics exhibited comparable performance across the three methods, the FPR95 metric indicated RNA's superiority, surpassing OE and PASCL by 4.55% and 6.11%, respectively. Furthermore, RNA demonstrats a 4.00% enhancement in ID classification accuracy compared to OE, and a 0.89% improvement over PASCL. These results validate RNA's effectiveness not only when trained with the 300K random images but also when employed with other OOD auxiliary sets.

## E.7 Structure of projection function

Table 15: OOD detection and ID classification performance (%) on CIFAR10-LT with various structure of projection function $h_\phi$. The gray regions indicate the results for our original proposed method (RNA).

| Training objective | AUC(↑) | AUPR(↑) | FPR95(↓) | ACC(↑) | Many(↑) | Medium(↑) | Few(↑) |
|---|---|---|---|---|---|---|---|
| $\|f_\theta(x)\|$ | 92.18 | 92.27 | 33.54 | 80.11 | 94.07 | 77.25 | 69.87 |
| $\|W_h^\top f_\theta(x)\|$ | 92.78 | 91.89 | 29.48 | 77.63 | 93.67 | 74.02 | 66.10 |
| $\|h_\phi(f_\theta(x))\|$ | 92.90 | 92.14 | 29.18 | 78.73 | 93.72 | 76.40 | 66.90 |

Table 16: The average norm of representations of ID training set (CIFAR10-LT), OOD training set (300K random images), ID test set (CIFAR10), and OOD test set (SVHN) with various structure of projection function $h_\phi$. The gray regions indicate the results for our original proposed method (RNA).

| Training objective | Norm of | ID train | OOD train | ID test | OOD test |
|---|---|---|---|---|---|
| $\|f_\theta(x)\|$ | | 79.89 | 7.93 | 59.31 | 4.61 |
| $\|W_h^\top f_\theta(x)\|$ | $\|f_\theta(x)\|$ | 66.29 | 2.59 | 46.32 | 1.30 |
| $\|h_\phi(f_\theta(x))\|$ | | 65.14 | 2.53 | 45.90 | 0.93 |

We conduct an ablation study to evaluate the impact of architectural variations, specifically focusing on the structure of the projection function $h_\phi$. In the proposed approach, we employ a 2-layer MLP configuration for $h_\phi$. As alternative configurations, we train the model using RNA loss on CIFAR10-LT, where in Equation 5, $\|h_\phi(f_\theta(x))\|$ is replaced either by the feature norm $\|f_\theta(x)\|$ or the linearly projected norm $\|W_h^\top f_\theta(x)\|$. The results presented in Table 15 indicate that the structure of the projection function $h_\phi$ is not a critical element in our method, as the OOD detection and ID classification performances remain comparable across the model variants. The reason we use the 2-layer MLP projection function in our proposed approach is to regulate the situations where the feature norm of ID samples increases excessively during training. Nevertheless, our findings indicate that employing the feature norm itself within the logarithmic term adequately prevents the divergence of $\|f_\theta(x)\|$, as shown in Table 16 and Section 5.2.

Table 17: The results of fine-tuning with auxiliary OOD data on pre-trained models by CIFAR10-LT, CIFAR100-LT, and ImageNet-LT. "PT"/"FT" refer to pre-training and fine-tuning, resp., where the fine-tuning loss is $\mathcal{L}_{\text{ID}} + \lambda\mathcal{L}_{\text{OOD}}$.

| PT CE | PT LA | FT ($\mathcal{L}_{\text{ID}}$) CE | FT ($\mathcal{L}_{\text{ID}}$) LA | FT ($\mathcal{L}_{\text{OOD}}$) | AUC (↑) | AUPR (↑) | FPR95 (↓) | ACC (↑) |
|---|---|---|---|---|---|---|---|---|
| | | | | ID dataset: CIFAR10-LT | | | | |
| ✓ | | ✓ | | OE | 90.23 | 90.15 | 39.51 | 76.82 |
| ✓ | | | ✓ | OE | 90.36 | 90.05 | 36.70 | 76.26 |
| | ✓ | | ✓ | | 90.29 | 89.89 | **36.69** | 76.21 |
| ✓ | | ✓ | | EnergyOE | 88.78 | 88.91 | 44.48 | 76.25 |
| ✓ | | | ✓ | EnergyOE | 85.10 | 83.36 | 49.01 | 70.28 |
| | ✓ | | ✓ | | 87.08 | 86.73 | 48.97 | 71.45 |
| ✓ | | ✓ | | RNA (ours) | 90.34 | 90.56 | 39.87 | 74.28 |
| ✓ | | | ✓ | RNA (ours) | 90.52 | 90.74 | 39.39 | **78.78** |
| | ✓ | | ✓ | | **90.53** | **90.85** | 36.70 | 78.53 |
| | | | | ID dataset: CIFAR100-LT | | | | |
| ✓ | | ✓ | | OE | 71.26 | 66.82 | 72.58 | 40.27 |
| ✓ | | | ✓ | OE | 70.91 | 66.52 | 72.57 | 41.09 |
| | ✓ | | ✓ | | 71.82 | 67.27 | 71.08 | 41.06 |
| ✓ | | ✓ | | EnergyOE | 69.18 | 66.24 | 77.23 | 39.46 |
| ✓ | | | ✓ | EnergyOE | 68.73 | 65.93 | 78.14 | 38.32 |
| | ✓ | | ✓ | | 69.13 | 66.35 | 77.61 | 38.24 |
| ✓ | | ✓ | | RNA (ours) | **73.02** | **68.93** | **70.65** | 40.69 |
| ✓ | | | ✓ | RNA (ours) | 72.58 | 68.72 | 71.99 | **44.82** |
| | ✓ | | ✓ | | 72.56 | 68.85 | 71.67 | 44.65 |
| | | | | ID dataset: ImageNet-LT | | | | |
| ✓ | | ✓ | | OE | 63.55 | 65.34 | 89.63 | 32.28 |
| ✓ | | | ✓ | OE | 64.68 | 65.66 | 88.05 | 37.23 |
| | ✓ | | ✓ | | 63.80 | 65.45 | 89.17 | 35.78 |
| ✓ | | ✓ | | EnergyOE | 63.81 | 65.13 | 88.29 | 38.43 |
| ✓ | | | ✓ | EnergyOE | 47.63 | 46.77 | 92.82 | 31.79 |
| | ✓ | | ✓ | | 36.47 | 40.31 | 95.59 | 19.70 |
| ✓ | | ✓ | | RNA (ours) | 68.95 | 68.94 | 86.05 | 34.66 |
| ✓ | | | ✓ | RNA (ours) | 69.27 | 69.22 | 85.37 | **38.91** |
| | ✓ | | ✓ | | **72.20** | **71.45** | **83.25** | 37.66 |

## E.8   Fine-tuning task with auxiliary OOD data

In the fine-tuning task for OOD detection (Hendrycks et al., 2019a; Liu et al., 2020; Papadopoulos et al., 2021), a classification model is initially trained without the OOD set and then later fine-tuned with the OOD set to improve its OOD detection ability.

We evaluate the effectiveness of our training method when applied to the fine-tuning task. The pre-trained models are obtained by training a model on CIFAR10/100-LT and ImageNet-LT, respectively, using only the classification loss, either CE or LA. We then train this model for 10 (respectively 3) more epochs on CIFAR10/100-LT and 300K random images (respectively ImageNet-LT and ImageNet-Extra) with an objective of $\mathcal{L}_{\text{ID}} + \lambda\mathcal{L}_{\text{OOD}}$. Here, we use $\mathcal{L}_{\text{CE}}$ or $\mathcal{L}_{\text{LA}}$ as $\mathcal{L}_{\text{ID}}$, and $\mathcal{L}_{\text{OE}}$, $\mathcal{L}_{\text{EnergyOE}}$, or $\mathcal{L}_{\text{RNA}}$ as $\mathcal{L}_{\text{OOD}}$. Table 17 summarizes the results. "PT" and "FT" stand for pre-training and fine-tuning methods, respectively. We can oberve that RNA attains the most optimal performances in both OOD detection and ID classification. Notably, when the classification loss during pre-training or fine-tuning is LA, the classification accuracy of models fine-tuned using OE or EnergyOE is significantly lower than that of RNA. This discrepancy could be attributed to the inherent trade-offs between OOD detection and ID classification, arising from the application of OE as discussed in Section 5.1.

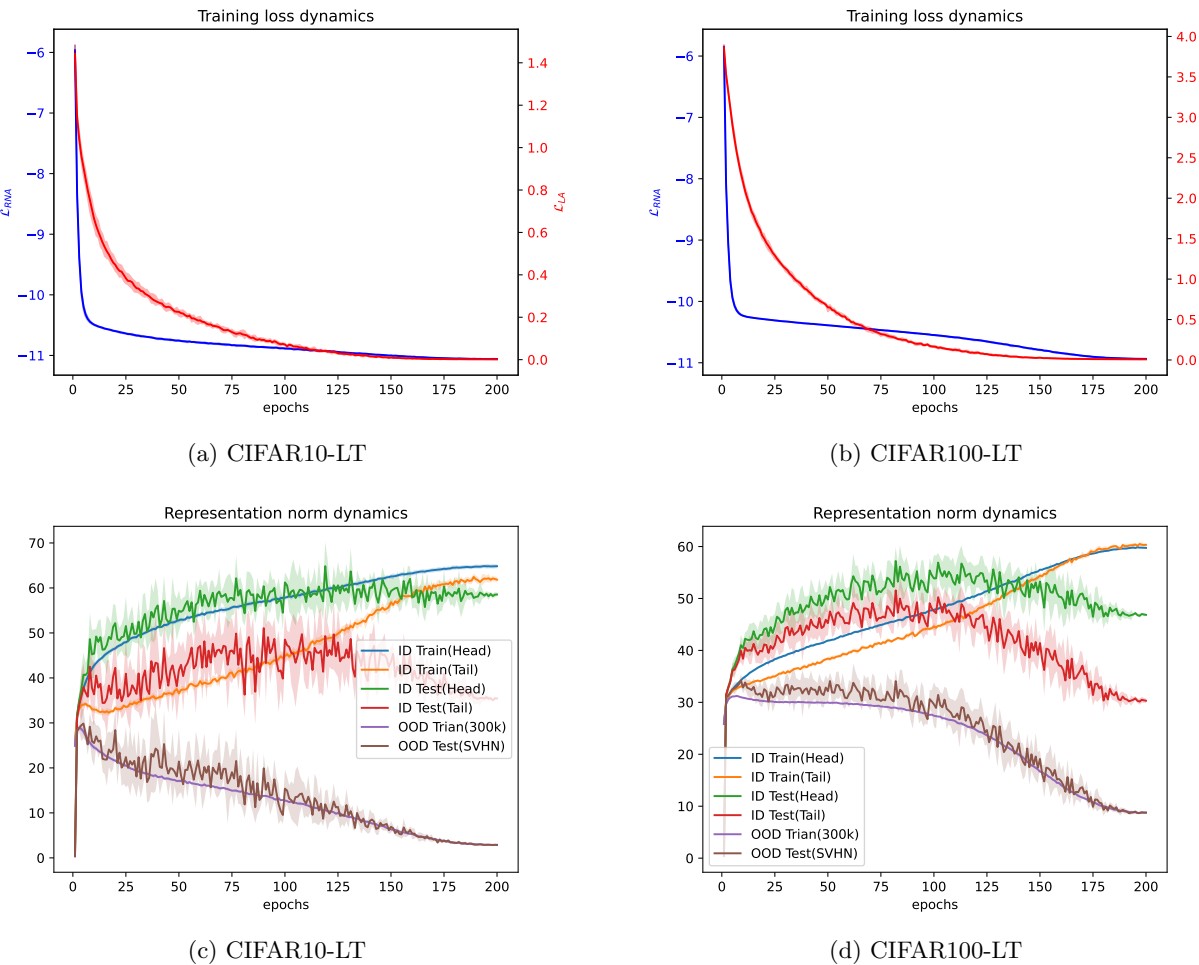

Figure 7: The training dynamics of the models trained with RNA on CIFAR10-LT and CIFAR100-LT. The training loss dynamics of LA loss (red) and RNA loss (blue) are in (a) and (b), and the representation norm dynamics of head and tail data in ID training and test dataset, OOD training set (300k random images), and OOD test set(SVHN) are presented in (c) and (d).

### E.9 Training dynamics of RNA

Figure 7a and 7b depict the changes of training losses (LA loss $\mathcal{L}_{LA}$ and RNA loss $\mathcal{L}_{RNA}$), respectively, over the training. Figure 7c and 7d illustrate the dynamics of representation norms of training ID, test ID, training OOD, and test OOD data, over the training. These figures demonstrate that the training effectively works to widen the gap between ID and OOD representation norms.

### E.10 Distributions of representation norms

We present additional figures similar to Figure 4b with different ID vs. OOD datasets in Figure 8. We observe a large gap between ID and OOD representation norms when CIFAR10 is used as the ID set, and a smaller but still distinct gap when CIFAR100 is the ID set, and a very small gap when CIFAR100 is the ID set and CIFAR10 is the OOD set, which correspond to the OOD detection performance results for each OOD test set in Table 8, Table 9, and Table 10.

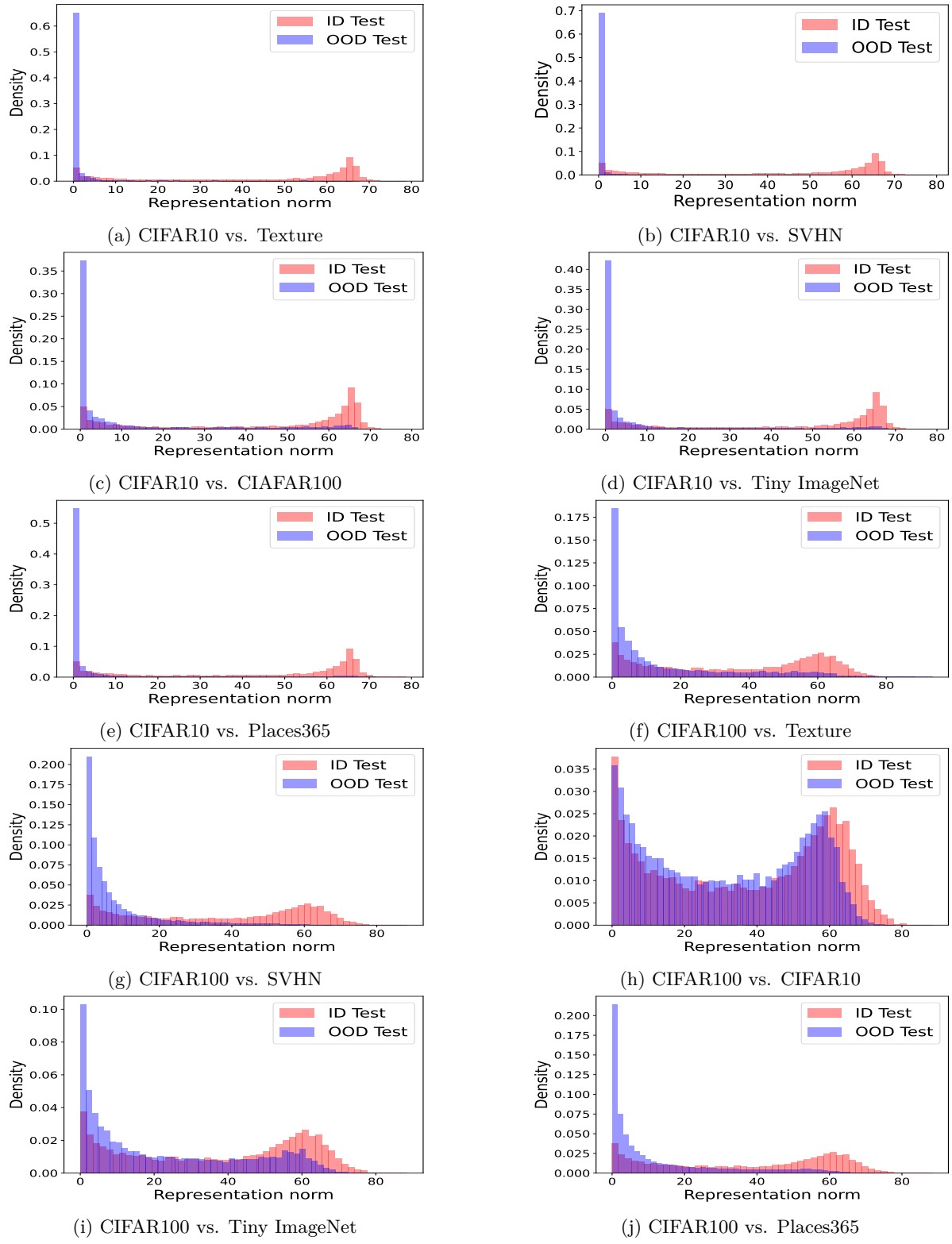

Figure 8: The histogram of representation norms of ID/OOD test data on RNA-trained models on CIFAR10-LT and CIFAR100-LT. The red bars represent the density of representation norms of ID data and the blue bars represent that of OOD data (Texture, SVHN, CIFAR, Tiny ImageNet, and Places365).

