# OpenReview forum: "Representation Norm Amplification for Out-of-Distribution Detection in Long-Tail Learning"
_TMLR — Accepted by TMLR_

### Review · Reviewer_aJme · 2024-06-21

**Summary Of Contributions:**

In their paper "Representation norm amplification for out-of-distribution detection in long-tail learning", the authors suggest a new method for training a classifier that simultaneously (i) performs well on rare classes; and (ii) allows out-of-distribution detection. The authors argue that existing methods show a trade-off between these two goals, whereas their new method achieves good performance on both metrics.

Caveat: I am not an expert in this field, am not familiar with the relevant literature, and am not sure why I was selected as a reviewer for this paper. So my confidence is low, but I nevertheless tried to provide a helpful review.

**Audience:**

Yes

**Broader Impact Concerns:**

None.

**Claims And Evidence:**

Yes

**Requested Changes:**

MAJOR COMMENTS

* I started reading the paper without knowing what "long-tail learning" means, what is "outlier exposure" or "logit adjustement". It took me some time to figure out what is going on. I'd recommend to more explicitly define "long-tail learning" and "long-tailed datasets" in the beginning. More importantly, it would be good to have a Background section in the beginning that would introduce OE and LA with all the relevant formulas. Currently OE is introduced in Section 4.1, after Table 1 is presented and discussed, and the equation for LA only appears several pages later in Section 4.2. The text would be clearer if all of that is introduced before Section 3.

* You say in Section 4.2 (page 6) that scaling the norm of ||f(x)|| does not affect class prediction. That's correct. But it does affect the logits and hence the softmax probabilities (which you don't say very clearly). You want OOD samples to have small norm, but small norm implies more uniform probabilities -- which is precisely the goal of OE (uniform probabilities for OOD samples), which you have criticized on top of the same page 6. I think you could make it more explicit here, why what you are doing is different.

* In Eq (3) you have ||h(f(x))|| instead of ||f(x)||, where h() is a 2-layer MLP. It was completely unclear to me why you used h() there at all -- this should be better motivated, and I would like to see an experiment where you remove h(). Does the method perform worse then? Why? In the text you always talk about the norm of f(x), not of some MLP of f(x).

* Fig 4b -- does this show the norms of f(x) or of h(f(x))? See previous question.

* It took me some time to understand the argument on top of page 8 about how OOD samples have any effect during training if they are ONLY used to update the BN layers. Here is my understanding: you are saying that BN+ReLU will lead to half of the data producing a representation with near-zero norm. But your loss encourages high norm (on ID data). This means that OOD data will _have_ to have near-zero norm. Is that the argument? Given that it's very non-intuitive and suprising, I suggest you make this argument more clear / more explicit. Also: Fig 4b is really striking; do you observe the same with all different ID and OOD datasets? Could you show the distributions? Also: does your method require that OOD sample size is as large as ID sample size? Maybe comment on that.


MINOR COMMENTS

* Eq 7 -- please explain the rationale behind LA in more detail, for those who are unfamiliar with it.

* Section 5.3 -- it's my terminology pet peeve, but "ablation" literally means "turning something off". If you are trying out different loss functions, it's not an "ablation study", it's "sensitivity analysis" or "hyperparameter choice" or something. Ablation study would be e.g. to remove the MLP h() from the loss.

**Strengths And Weaknesses:**

Strengths: The paper is easy to follow. It critically evaluates existing methods, suggests a new method, and provides a comprehensive evaluation. The suggestion is straightforward but nontrivial and is was actually very surprising to me that it works at all.

Weaknesses: Introduction/background could use reorganizing to improve clarity for non-experts.

Overall, it seems a very good fit for TMLR and can be accepted after minor revisions. However, note that I am not an expert and so my confidence is low.

---

> ### Author Response · Authors · 2024-07-06
> **Response to Reviewer aJme (1/2)**
>
> We sincerely appreciate the reviewer for the valuable feedback. Please find our response below.
>
> **1. Background section**
> >(Major comment 1 & Minor comment 1)  ...it would be good to have a Background section in the beginning that would introduce OE and LA with all the relevant formulas.
>
> We have added the Background section that introduces long-tailed datasets, long-tail learning, OE, and LA.
>
> **2. Handling Underconfidence issue in OE with RNA(ours)**
> >(Major comment 2) You say in Section 4.2 (page 6) that scaling the norm of $||f(x)||$ does not affect class prediction. That's correct. But it does affect the logits and hence the softmax probabilities (which you don't say very clearly). You want OOD samples to have small norm, but small norm implies more uniform probabilities -- which is precisely the goal of OE (uniform probabilities for OOD samples), which you have criticized on top of the same page 6. I think you could make it more explicit here, why what you are doing is different.
>
> While it is true that a small representation norm implies a uniform distribution of softmax probability, OE training induces the underconfidence in tail data during training, because regulating rare OOD samples adversely affects the tail samples. In contrast, our proposed RNA enlarges the norm of ID representations including both head and tail samples, leading to higher confidence levels for ID samples. Additionally, RNA training does not perturb classifier weights with OOD samples while OE training involves OOD representations in the classifier weight updates.
>
> **3. Ablation study for $h$ function**
> >(Major comment 3) It was completely unclear to me why you used $h()$ there at all.
>
> We present related experimental results in Appendix E.5. We evaluate RNA without $h$, with linear $h$ function, and with original two-layer MLP. The rationale for incorporating $h$ is to regulate the situations where the representation norm of ID samples increases excessively during training. However, our results indicate that employing the feature norm itself within the logarithmic term adequately prevents the divergence of the norm of $f(x)$, as shown in Table 14.
>
> **4. Fig 4b explanation**
> >(Major comment 4) Fig 4b -- does this show the norms of $f(x)$ or of $h(f(x))$?
>
> The figure 4b shows the norms of $f(x)$ rather than $h(f(x))$.

---

> ### Author Response · Authors · 2024-07-06
> **Response to Reviewer aJme (2/2)**
>
> **5. The rationale behind RNA training with BN+ReLU**
> >(Major comment 5-1) It took me some time to understand the argument on top of page 8 about how OOD samples have any effect during training if they are ONLY used to update the BN layers. Here is my understanding: you are saying that BN+ReLU will lead to half of the data producing a representation with near-zero norm. But your loss encourages high norm (on ID data). This means that OOD data will have to have near-zero norm. Is that the argument? Given that it's very non-intuitive and suprising, I suggest you make this argument more clear / more explicit.
>
> BN+ReLU leads to a portion of the values (not exactly half, as shown in Figure 2b) being zero at each coordinate of the representation vectors, rather than a portion of the data to be near-zero norm. It is because the BN layer normalizes each coordinate separately. However, our loss encourages high values in the vector coordinates of ID data, so OOD data have near-zero norms and ID data have high representation norms.
>
> **6. Histograms of representation norms with different ID and OOD test sets**
> >(Major comment 5-2) Fig 4b is really striking; do you observe the same with all different ID and OOD datasets? Could you show the distributions?
>
> We present the figures in Appendix E.8 in the revised version. We observe the similar results with all different ID and OOD datasets except when ID is CIFAR100-LT and OOD is CIFAR10 or Tiny ImageNet. This results aligns to the OOD detection performance across different ID and OOD datasets in Appendix E.1, which implies that RNA has lower performance for near OOD tasks rather than far OOD tasks.
>
> **7. OOD sample size**
> >(Major comment 5-3) does your method require that OOD sample size is as large as ID sample size?
>
> In Appendix E.10 of the revised version, we present experiments varying the OOD sample size. Our method demonstrates worse FPR95 performances with smaller OOD sample sizes.
>
> Our approach detects OOD data based on representation norms, which are proportional to the gap between latent vectors before the last BN layer and the running mean of the last BN layer. So, we will check the gaps with varying the OOD sample size.
>
> Let $\mu_{\text{ID}}$ and $\mu_{\text{OOD}}$ denote the mean of the latent vectors before the last BN layer for ID and OOD samples, respectively, and $\mu_{\text{BN}}$ denote the running mean of the last BN layer. After training with RNA, we observe that $\mu_{\text{ID}}>\mu_{\text{BN}}>\mu_{\text{OOD}}$, as shown in Figure 3b, which implies that many ID representations are activated, while many OOD representations are deactivated after RNA training.
>
> For superior OOD detection based on representation norms, ID representation norms should be relatively larger than OOD representation norms. Thus, $\frac{\mu_{\text{ID}}-\mu_{\text{BN}}}{\mu_{\text{BN}}-\mu_{\text{OOD}}}$ is proportional to the OOD detection performance. At different ratio of ID to OOD, different number of ID or OOD samples accounts for calculating $\mu_{\text{BN}}$, since $\mu_{\text{BN}}$ is the mean of all the ID and OOD samples in the batch. Therefore, as the OOD sample size decreases, $\mu_{\text{BN}}$ gets closer to $\mu_{\text{ID}}$ than $\mu_{\text{OOD}}$, so the ratio $\frac{\mu_{\text{ID}}-\mu_{\text{BN}}}{\mu_{\text{BN}}-\mu_{\text{OOD}}}$ decreases, as shown in Table 17 on page 30 of the revised paper. This decreased ratio $\frac{\mu_{\text{ID}}-\mu_{\text{BN}}}{\mu_{\text{BN}}-\mu_{\text{OOD}}}$ leads to smaller gap of representation norms between ID and OOD and declined OOD detection performance.
>
> **8. Renaming Section Ablation study**
>
> We have renamed the section “Ablation study” to “Additional experiments” in the revised version.

---

> > ### Comment · Reviewer_aJme · 2024-07-08
> > **Thank you**
> >
> > Thanks for your edits.
> >
> > I am almost satisfied, but some of the appendices that you added now do not seem to be references in the main text. E.g. E.8 does not seem to be referenced anywhere at all (searching for "E.8" only gives one hit: the title of section E.8). Same for E.9 and E.10. Please make sure that every single appendix section is referenced somewhere in the main text with a brief explanation of what's inside.
> >
> > Also, there is a typo in Fig 8h caption.

---

> > > ### Author Response · Authors · 2024-07-08
> > > **Response to Reviewer aJme**
> > >
> > > We are pleased that our rebuttal has addressed most of your concerns.
> > >
> > > In the revised paper, we have added a reference for the RN histograms on other pairs of ID and OOD test sets on the page 9 in blue color.
> > > We have also added the references for the additional experiments on the size of auxiliary OOD sets and the training batch ratio of ID and OOD samples on the page 11 of the revised paper in blue color.
> > >
> > > Also, we have reordered the subsections related to auxiliary OOD sets as follows:
> > >
> > >   - "Size of auxiliary OOD sets" (E.9 $\rightarrow$ E.4)
> > >
> > >   - "Training batch ratio of ID and OOD samples" (E.10 $\rightarrow$ E.5)
> > >
> > > We have corrected the typo in Fig 8h caption.
> > >
> > > We sincerely appreciate your support for our work, and please let us know if you have any further questions.

---

### Review · Reviewer_gNpe · 2024-06-24

**Summary Of Contributions:**

This paper is proposed to address the problem, the conflict between OOD detection and long-tailed classification. The authors analysed in detail the contribution of the Logit Adjustment loss, which prefers the long-tailed classification, and the Outlier Exposure loss, which prefers the OOD detection, to the classifier weights, and observed the bad impact of the OE loss on the weights of the tailed classes, which would affect the performance of the ID samples. Therefore, they propose a novel method called Representation Norm Amplification, which disentangles ID classification and OOD detection in long-tail learning by detecting the norm of the samples' feature to find the OOD samples. The experiments conducting on CIFAR and ImageNet datasets demonstrate the effectiveness of the proposed method especially for the tailed classes.

**Audience:**

Yes

**Broader Impact Concerns:**

The work is proposed to tackle the challenging scenarios where reliable OOD detection needs to be performed on models trained with long-tailed distributions. This is a crucial and necessary safety issue that neural networks need to address before being deployed in the real world.

**Claims And Evidence:**

Yes

**Requested Changes:**

1. Further discussion and analysis is required as to why it is optimal to perform RNA loss only on ID samples, with no restrictions on OOD samples.
2. Sensitive analysis of the ratio of ID samples and OOD samples within each batch is required.

**Strengths And Weaknesses:**

**Strengths:**

1. The logic of the whole article is clear and well-written.
2. The observation of the trade-offs between OOD detection and long-tailed recognition and motivation sound reasonable. The proposed solution in this paper is just right and fits the motivation.
3. The experiments presented in the manuscript are valuable especially for the ablation study.

**Weaknesses or Questions:**

1. Although the authors explain in the methodology section that the truncation effect of the ReLU is helpful in detecting the OOD samples, I wonder why the results of reducing the representation norm of the OOD samples would be lower than only performing above ID samples.
2. Since the success in detecting OOD samples in the manuscript contributes to the ReLU truncation effect and RNA loss, I don't know if the performance of the method is sensitive to the ratio of ID samples to OOD samples within each batch.

---

> ### Author Response · Authors · 2024-07-06
> **Response to Reviewer gNpe**
>
> We appreciate the constructive feedback and carefully address the comments.
>
> **1. RNA and the OOD attenuation loss**
> > (Requested change 1) Further discussion and analysis is required as to why it is optimal to perform RNA loss only on ID samples, with no restrictions on OOD samples.
>
> We conjecture that the reason training with $\mathcal{L}_ {\text{ID-amp.}} + \mathcal{L}_ {\text{OOD-att.}}$ underperforms compared to RNA in OOD detection is that $\mathcal{L}_ {\text{OOD-att}}$ causes the OOD representations to converge to the null space of the $h_\phi$ function. Although the OOD attenuation loss does not guarantee this convergence, training with OOD attenuation loss appears to result in this scenario in our experiments, as $||h_\phi(f_\theta(x_\text{OOD}))||$ approaches to zero while $||f_\theta(x_\text{OOD})||$ does not. In contrast, RNA training does not explicitly move the OOD representations towards the null space of $h_\phi$, but instead indirectly attenuates them through the last BN+ReLU layers. Furthermore, $\mathcal{L}_ {\text{ID-amp.}} + \mathcal{L}_ {\text{OOD-att.}}$ deteriorates ID classification performance compared to RNA. Therefore, directly attenuating OOD representations is not appropriate in the LT-OOD task.
>
> **2. The batch ratio of ID and OOD samples**
> >(Requested change 2) Sensitive analysis of the ratio of ID samples and OOD samples within each batch is required.
>
> In Appendix E.10 of the revised version, we present experiments varying the OOD sample size. Our method demonstrates worse FPR95 performances with smaller OOD sample sizes.
>
> Our approach detects OOD data based on representation norms, which are proportional to the gap between latent vectors before the last BN layer and the running mean of the last BN layer. So, we will check the gaps with varying the OOD sample size.
>
> Let $\mu_{\text{ID}}$ and $\mu_{\text{OOD}}$ denote the mean of the latent vectors before the last BN layer for ID and OOD samples, respectively, and $\mu_{\text{BN}}$ denote the running mean of the last BN layer. After training with RNA, we observe that $\mu_{\text{ID}}>\mu_{\text{BN}}>\mu_{\text{OOD}}$, as shown in Figure 3b, which implies that many ID representations are activated, while many OOD representations are deactivated after RNA training.
>
> For superior OOD detection based on representation norms, ID representation norms should be relatively larger than OOD representation norms. Thus, $\frac{\mu_{\text{ID}}-\mu_{\text{BN}}}{\mu_{\text{BN}}-\mu_{\text{OOD}}}$ is proportional to the OOD detection performance. At different ratio of ID to OOD, different number of ID or OOD samples accounts for calculating $\mu_{\text{BN}}$, since $\mu_{\text{BN}}$ is the mean of all the ID and OOD samples in the batch. Therefore, as the OOD sample size decreases, $\mu_{\text{BN}}$ gets closer to $\mu_{\text{ID}}$ than $\mu_{\text{OOD}}$, so the ratio $\frac{\mu_{\text{ID}}-\mu_{\text{BN}}}{\mu_{\text{BN}}-\mu_{\text{OOD}}}$ decreases, as shown in Table 17 on page 30 of the revised paper. This decreased ratio $\frac{\mu_{\text{ID}}-\mu_{\text{BN}}}{\mu_{\text{BN}}-\mu_{\text{OOD}}}$ leads to smaller gap of representation norms between ID and OOD and declined OOD detection performance.

---

> > ### Comment · Reviewer_gNpe · 2024-07-25
> > **Response to Authors**
> >
> > Thank you for your reply. My concerns have been addressed.

---

> > > ### Author Response · Authors · 2024-07-25
> > > **Response to Reviewer gNpe**
> > >
> > > We are pleased that our rebuttal successfully addressed your concerns.
> > > We sincerely appreciate your valuable feedback again, and please let us know if you have any further questions.

---

### Review · Reviewer_LZ9f · 2024-06-25

**Summary Of Contributions:**

This paper aims to solve the OOD detection in long-tail learning and focuses on the balance between OOD detection and long-tail recognition. The authors propose Representation Norm Amplification (RNA) to solve this challenge. The main contributions are as follows:
1. Utilizing the norm of the representation as a new dimension for OOD detection, thus generating a noticeable discrepancy in the representation norm between ID and OOD data during the training process without perturbing the feature learning for ID classification.
2. The authors conduct a series of experiments and validate the effectiveness of the proposed method.

**Audience:**

Yes

**Claims And Evidence:**

No

**Requested Changes:**

1. The authors need a more detailed explanation of the claim that ''The OOD data does not contribute to model parameter updates''.
2. The proposed idea is general for OOD detection task. It is not specifically designed for long-tail recognition. The authors should provide extra explanation and clarify this.

**Strengths And Weaknesses:**

Pros:
1. This paper is well-organized and easy to follow.
2. The authors are inspired by the trade-off between OOD detection and long-tailed recognition which is well-motivated and indeed novel.
3. The experimental results demonstrate the superior performance of RNA compared to other baseline approaches across various OOD test sets.

Cons:
1. The claim that ''The OOD data does not contribute to model parameter updates'' is wrong.
2. The authors mentioned calibration in Sec. 5.3.5. However, ECE is not a ''perfect'' metric for the task. It is non-differentiable. Other metrics should be discussed here.

---

> ### Author Response · Authors · 2024-07-06
> **Response to Reviewer LZ9f**
>
> We sincerely thank the reviewer for the constructive feedback. Please find our response below.
>
> **1. OOD data for model updates**
> > (Con 1 & Requested change 1) The authors need a more detailed explanation of the claim that ''The OOD data does not contribute to model parameter updates''.
>
> As noted by the reviewer, OOD data are indeed involved in the updates of BN parameters, but they are involved only indirectly. The BN parameters updated by both ID and OOD influence the latent vectors of ID data. However, OOD data do not contribute to the gradients of the loss, as the loss is computed solely with ID data. Also, the BN statistics that involve OOD representations are not part of the backpropagation computation graph and are updated using a moving average rather than gradient descent. Therefore, the gradient updates are not contributed by the OOD data. In conclusion, we acknowledge that our claim was too strong, and have revised the statement ''The OOD data does not contribute to model parameter updates'' into "The OOD data only indirectly influence the updating of the model parameters" in the caption of Figure 3 of the revised paper.
>
> **2. Calibration performance (NLL)**
> >(Cons 2) The authors mentioned calibration in Sec. 5.3.5. However, ECE is not a ''perfect'' metric for the task. It is non-differentiable. Other metrics should be discussed here.
>
> We demonstrate calibration performance using Negative Log-Likelihood (NLL), a differentiable metric defined as $-\frac{1}{N}\sum^N_{i=1}\log(p_{i, y_i})$, where lower values indicate better calibration. Our proposed RNA outperforms CE, OE, and PASCL in terms of NLL on CIFAR10, CIFAR100, and ImageNet.
>
> | Method | CIFAR10 | CIFAR100 | ImageNet |
> |---|:---:|:---:|:---:|
> | CE | 1.46 | 3.13 | 3.19 |
> | OE | 1.17 | 3.30 | 3.19 |
> | PASCL | 1.14 | 3.34 | 3.27 |
> | RNA | 0.89 | 2.65 | 2.55 |
>
> **3. RNA and the data imbalance**
> >(Requested change 2) The proposed idea is general for OOD detection task. It is not specifically designed for long-tail recognition. The authors should provide extra explanation and clarify this.
>
> Our method does not explicitly account for tail data in the training loss function or the OOD scoring method. However, we address the inherent trade-offs between long-tail learning and OOD detection, which appear when controlling confidence for both tasks. To this end, we propose a training and scoring method that decouples long-tail learning and OOD detection. Our approach effectively distinguishes ID and OOD representation norms without controlling confidence, while adjusting for the bias between head and tail data by controlling confidence. This strategy demonstrates superior performance in both OOD detection and ID classification.
>
> Additionally, our method is applicable to OOD detection in balanced settings. We present the experimental results for various imbalance ratios of the training dataset, including the balanced training set, i.e. $\rho=1$, in Table 5 on page 5 of the main paper.
>
> In conclusion, our proposed method tackles long-tailed OOD detection task by decoupling OOD detection and ID classification and is also effective for OOD detection in balanced settings.

---

> > ### Comment · Reviewer_LZ9f · 2024-07-15
> > **Response to Authors**
> >
> > I think issue 1 and 3 are adequately addressed. However, for issue 2, I would like to see more recent metrics such as KSE [1] and others.
> >
> > [1]Gupta, Kartik, et al. "Calibration of Neural Networks using Splines." International Conference on Learning Representations.

---

> > > ### Author Response · Authors · 2024-07-17
> > > **Response to Reviewer LZ9f**
> > >
> > > Thank you for your additional comments.
> > >
> > > We present the KSE performance (%) in the table below, with lower values indicating better performance. Our proposed method, RNA, achieves the best KSE performance compared to CE, OE, and PASCL on CIFAR10, and the second-best on CIFAR100 and ImageNet.
> > >
> > > | Method | CIFAR10 | CIFAR100 | ImageNet |
> > > |---|---|---|---|
> > > | CE | 20.51 | 32.12 | 21.55 |
> > > | OE | 13.67 | 11.78 | 14.09 |
> > > | PASCL | 14.37 | 14.00 | 12.43 |
> > > | RNA (ours) | 7.22 | 12.18 | 12.64 |
> > >
> > > We sincerely appreciate your support for our work. Please feel free to let us know if you have any further questions.

---

### Decision · Action_Editor_Db9G · 2024-08-16

**Recommendation:** Accept as is

**Comment:**

There was a consensus among the reviewers that the paper not only supports its claims with clear and convincing evidence but is of interest to the TMLR community.

The paper also benefited from the review process with the addition of some further background sections at the start of the paper as requested by reviewer [aJme].

For my part, I'd encourage the authors to lookup some of the literature on 'evidential networks' such as: https://arxiv.org/abs/2311.11367, https://arxiv.org/abs/1806.01768, and https://arxiv.org/abs/1802.10501, which I think is relevant to the approach here.  Those works try to reinterpret the output of a classifier network as the parameters of a Dirichlet distribution rather than a Categorical.  There the norm of the output has a direct operational meaning tied with the certainty of the prediction.

Given the consensus among reviewers and the clear audience appeal, I'll accept as is.

**Audience:**

The paper addresses out of distribution detection, a problem of interest to the TMLR community.

**Claims And Evidence:**

This paper suggests a mechanism for out of distribution detection.  Add an auxillary loss that attempts to grow the norm of the output for in distribution examples.  They have analysis and experiments that suggest that this can work well.

All reviewers agreed that the paper provided clear and convincing evidence of the claims it makes.